# CORDS: Continuous Representations of Discrete Structures

**Tin Hadži Veljković**
UvA-Bosch Delta Lab
University of Amsterdam

**Erik Bekkers**
AMLab
University of Amsterdam

**Michael Tiemann**[*]
Tübingen AI Center
Eberhard Karls Universität Tübingen

**Jan-Willem van de Meent**
UvA-Bosch Delta Lab
University of Amsterdam

## Abstract

Many learning problems require predicting sets of objects when the number of objects is not known beforehand. Examples include object detection, molecular modeling, and scientific inference tasks such as astrophysical source detection. Existing methods often rely on padded representations or must explicitly infer the set size, which often poses challenges. We present a novel strategy for addressing this challenge by casting prediction of variable-sized sets as a continuous inference problem. Our approach, CORDS (*Continuous Representations of Discrete Structures*), provides an invertible mapping that transforms a set of spatial objects into continuous fields: a density field that encodes object locations and count, and a feature field that carries their attributes over the same support. Because the mapping is invertible, models operate entirely in field space while remaining exactly decodable to discrete sets. We evaluate CORDS across molecular generation and regression, object detection, simulation-based inference, and a mathematical task involving recovery of local maxima, demonstrating robust handling of unknown set sizes with competitive accuracy.

## 1 Introduction

In problems where we wish to reason about discrete structure, we often need to reason about a set of objects without knowing the cardinality of this set in advance. Examples include object detection (Tian et al., 2019; Wang et al., 2024a), scientific inference tasks, such as reconstructing catalogs of astrophysical sources (Vafaei Sadr et al., 2019; Cornu et al., 2024), or conditional molecular generation, where the conditioned property does not uniquely determine the number of atoms (Faltings et al., 2025; Pham et al., 2022). Inferring cardinality directly from data is often difficult, making sampling in conditional generation or inference tasks inefficient.

Reasoning about unknown cardinality is a challenge that has been around for a long time. Classic approaches include model selection with variational inference (Beal, 2003), reversible jump MCMC (Richardson & Green, 1997), and Bayesian nonparametrics (Hjort et al., 2010). In modern approaches based on deep learning, a common strategy is to *pre-allocate capacity* beyond what is typically needed, and then suppress or ignore unneeded capacity (Xu et al., 2024). In the sciences, similar ideas appear when combining variational inference and empirical Bayes estimation, where learning the prior can serve to prune unneeded degrees of freedom (van de Meent et al., 2014). These examples reflect a pervasive pattern: rather than modeling the distribution over cardinalities $p(N)$ explicitly, many methods sidestep the issue by way of user-specified truncations or paddings.

In parallel, *continuous representations* have become increasingly common across domains, offering flexible ways to encode signals and structures and partly addressing the challenges of variable cardinality. Neural fields and coordinate-based models (Mildenhall et al., 2020; Sitzmann et al., 2020;

---

[*]Part of this work was carried out when Michael was still with the Bosch Center for Artificial Intelligence, Renningen, Germany.

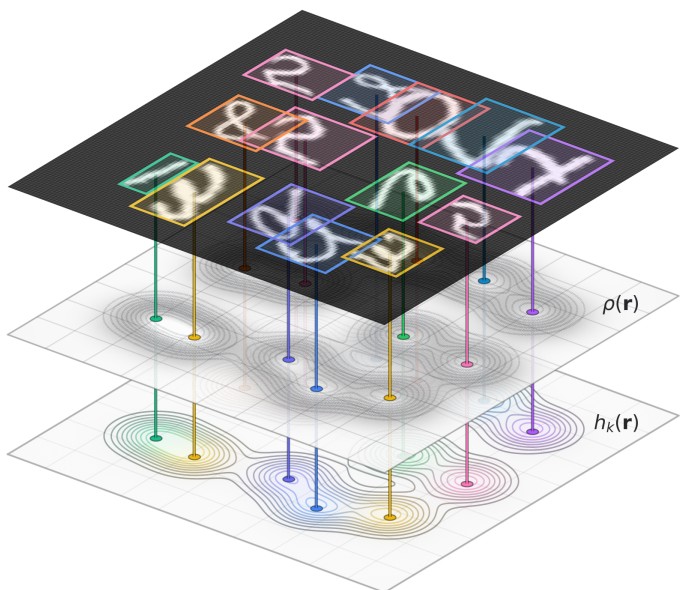

Figure 1: An image with $N$ MNIST digits (top) is encoded with CORDS into a density field $\rho(\mathbf{r})$ (middle) and per-class feature fields $h_k(\mathbf{r})$ (bottom). The number of objects is encoded directly in the density mass, $N = \int \rho(\mathbf{r})\, d\mathbf{r}$.

Xie et al., 2022) showed how images and scenes can be embedded in continuous domains, and this perspective has since been extended to molecules and proteins Pinheiro et al. (2024); Kirchmeyer et al. (2025); Faltings et al. (2025). These approaches remove the need to specify the number of objects in advance, since atoms or components are represented as smooth densities that can, in principle, be sampled or inpainted flexibly Faltings et al. (2025). Yet cardinality is still only inferred indirectly, and object attributes are often added afterwards through auxiliary classifiers or peak-detection heuristics, rather than being built into the representation itself. As a result, continuous fields provide flexibility but not a unified treatment of both counts and features.

To address this gap, we introduce CORDS (*Continuous Representations of Discrete Structures*). Here, discrete objects are mapped into *continuous fields*, smooth functions defined over an ambient domain (e.g. space, time, or grids), where a density field encodes object counts and positions, and a feature field carries their attributes. The total density mass then acts as a continuous, differentiable quantity that implicitly encodes the number of objects. The construction is bijective, so models can operate directly in field space and still recover discrete sets without auxiliary mechanisms such as fixed slots or padding. This provides a systematic alternative to existing workarounds, offering a single representation that applies across domains.

## 2 RELATED WORK

**Generative modeling on graphs.** Graph generative modeling is typically framed in either discrete or continuous terms. Discrete methods treat molecules as graphs of nodes and edges encoded through adjacency matrices, as in DiGress (Vignac et al., 2023), which applies discrete denoising diffusion, variational flow matching (Eijkelboom et al., 2024), which casts flow matching as variational inference over categorical states, and MoFlow (Zang & Wang, 2020), which uses invertible flows to generate atoms and bonds with exact likelihoods. Continuous methods instead generate atomic coordinates and features directly in 3D space. Examples include G-SchNet (Gebauer et al., 2020), which models molecular conformations with equivariant GNNs, ENF (Satorras et al., 2022), which extends normalizing flows with equivariant dynamics, and diffusion-based approaches such as EDM (Hoogeboom et al., 2022), EDM-Bridge (Wu et al., 2022), and GeoLDM (Xu et al., 2023). More recent architectures such as Ponita (Bekkers et al., 2024) and Rapidash (Vadgama et al., 2025) further improve scalability with multi-scale equivariant representations.

Alongside these approaches, several works have explored representing molecular graphs in continuous fields rather than explicit graphs. VoxMol (Pinheiro et al., 2024) represents molecules as voxelized density grids processed by convolutional networks, while Ragoza et al. (Ragoza et al., 2020) introduced one of the earliest 3D molecular generative models based on atomic density fields, reconstructing molecules through peak detection and bond heuristics. FuncMol (Kirchmeyer et al., 2025) proposed a neural field parameterization of molecular occupancy, and ProxelGen (Faltings et al., 2025) extended this idea to conditional generation and inpainting. While these methods remove the need to fix graph size in advance, they ultimately recover atoms and features through thresholding or auxiliary classifiers, leaving cardinality and features only indirectly modeled.

**Object detection.** Two-stage detectors such as Faster R-CNN (Ren et al., 2016) generate region proposals followed by refinement, while one-stage models like YOLO (Wang et al., 2024a) and RetinaNet (Lin et al., 2018) predict boxes and classes densely, balancing speed and accuracy. EfficientDet (Tan et al., 2020) further improves this trade-off via compound scaling. Transformer-based approaches, exemplified by Deformable DETR (Zhu et al., 2021) and the real-time RT-DETR family (Wang et al., 2024b), accelerate convergence and improve small-object handling. Methods that operate through heatmaps or density maps, e.g., CenterNet (Duan et al., 2019), crowd counting (Xu et al., 2019), and microscopy detection (Li et al., 2022), are most closely related to our setting, as they localize objects from continuous spatial representations. However, unlike CORDS, these approaches focus purely on localization and do not model or recover object-level features from the underlying fields.

**Simulation-based inference.** Simulation-based inference (SBI) is used in domains such as cosmology, astrophysics, and particle physics where the likelihood is intractable but simulators are available. Early approaches such as Approximate Bayesian Computation (ABC) (Beaumont et al., 2002) relied on handcrafted summary statistics, while modern neural methods—neural posterior estimation (NPE) and neural ratio estimation (NRE) (Papamakarios et al., 2019; Lueckmann et al., 2021)—provide flexible and scalable inference. Recent advances include flow matching posterior estimation (FMPE) (Dax et al., 2023), which leverages flow matching to improve scalability and accuracy, achieving state-of-the-art results in gravitational-wave inference. Yet, as in detection, variable event cardinalities are still typically handled by padding, rather than modeled directly.

We provide a more exhaustive survey of related work in Appendix E.

## 3 CORDS: Continuous fields for variable-size sets

Our goal is to establish a *bijective* correspondence between discrete sets and continuous fields. This allows models to operate directly in the field domain, where learning and generation are often more convenient, while still ensuring that discrete predictions can be recovered exactly whenever needed. The construction applies uniformly across modalities: the only difference lies in the choice of the ambient domain $\Omega \subseteq \mathbb{R}^d$ (e.g. a pixel grid for images, three-dimensional space for molecules, or the time axis for light curves).

We consider a set $S = \{(\mathbf{r}_i, \mathbf{x}_i)\}_{i=1}^N$ of objects with positions $\mathbf{r}_i \in \Omega$ and feature vectors $\mathbf{x}_i \in \mathbb{R}^{d_x}$. Let $K : \Omega \times \Omega \to \mathbb{R}_{\geq 0}$ be a continuous, positive kernel with finite, location-independent mass $\alpha = \int_\Omega K(\mathbf{r}; \mathbf{s}) \, d\mathbf{r}$.

**Encoding.** In the CORDS approach, a discrete set is transformed into continuous fields by *superimposing kernels* centered at the object positions. The resulting density field represents where objects are located, while the feature field aligns with it by distributing the object attributes over the same spatial support:

$$\rho(\mathbf{r}) \;=\; \frac{1}{\alpha} \sum_{i=1}^N K(\mathbf{r}; \mathbf{r}_i), \qquad \mathbf{h}(\mathbf{r}) \;=\; \frac{1}{\alpha} \sum_{i=1}^N \mathbf{x}_i \, K(\mathbf{r}; \mathbf{r}_i). \tag{1}$$

Our next objective is to establish conditions for exact invertibility so that $(\rho, \mathbf{h})$ determine the set uniquely.

**Decoding.** The inversion is made possible by three structural properties of the encoding. The total mass of the density determines the number of objects, since each kernel contributes the same constant integral $\alpha$. The shape of the density identifies object locations, as it must be explained by a superposition of kernel translates. Finally, the feature field is aligned with the density, so projecting it onto the recovered kernels yields the original features. We now formalize each of these steps.

1. *Cardinality.* Each object is represented via a kernel whose integral is $\alpha$, so that

$$N \; = \; \int_\Omega \rho(\mathbf{r}) \, d\mathbf{r}. \tag{2}$$

   This makes variable cardinality straightforward: the number of objects is inferred directly from the density field.

2. *Positions.* Once the number of objects is known, their locations are encoded in the shape of the density field. Because $\rho$ is by definition a superposition of kernel translates, positions can be recovered by solving the kernel-matching problem

$$\min_{\mathbf{r}_1, \ldots, \mathbf{r}_N} \int_\Omega \left( \rho(\mathbf{r}) - \frac{1}{\alpha} \sum_{i=1}^N K(\mathbf{r}; \mathbf{r}_i) \right)^2 d\mathbf{r}. \tag{3}$$

   If the field truly originates from the forward transformation, the original centers achieve the global minimum. In practice, approximate solutions found with gradient-based optimization already suffice, and can be further refined if higher accuracy is required.

3. *Features.* Once the positions are fixed, the final step is to recover the object features. Because the feature field was constructed from the same kernels as the density field, its support aligns with the recovered positions. For each position $\mathbf{r}_i$ we define $\kappa_i(\mathbf{r}) = K(\mathbf{r}; \mathbf{r}_i)$; these kernels span the subspace of the feature field, so reconstructing the features amounts to projecting $\mathbf{h}$ onto this basis. To make this concrete, we form the Gram matrix $G \in \mathbb{R}^{N \times N}$ with entries

$$G_{ij} = \int_\Omega \kappa_i(\mathbf{r}) \, \kappa_j(\mathbf{r}) \, d\mathbf{r},$$

   and the projection matrix $B \in \mathbb{R}^{N \times d_x}$ with rows

$$B_{i:} = \int_\Omega \mathbf{h}(\mathbf{r}) \, \kappa_i(\mathbf{r}) \, d\mathbf{r}.$$

   The system $B = \frac{1}{\alpha} G \mathbf{X}$ then recovers the feature matrix $\mathbf{X} \in \mathbb{R}^{N \times d_x}$, whose rows are the feature vectors $\mathbf{x}_i$. Under mild assumptions on the kernel, $G$ is symmetric positive-definite, ensuring that this system has a unique solution

$$\mathbf{X} = \alpha G^{-1} B. \tag{4}$$

   This solution exactly matches the features that generated the field during encoding.

With this construction, we obtain a bijection between finite sets and their corresponding fields. The conditions that guarantee exact recovery, together with the formal results and proofs, are detailed in Appendix A. In contrast, Appendix C.1 focuses on how we approximate decoding in practice.

**Relation to kernel mean embeddings.** Equation 1 is closely related to *kernel mean embeddings* (KMEs), which represent a distribution (or empirical sample) as a function given by a kernel-weighted superposition of its support points (Muandet et al., 2017). While KMEs are usually used to embed distributions for learning, CORDS uses the same kernel-superposition principle to represent finite sets of objects as continuous fields, with a constructive decoding back to the underlying set under mild identifiability assumptions.

### 3.1 PRACTICAL CONSIDERATIONS.

**Sampling strategies.** Fields are defined on a continuous domain $\Omega$, but training requires a finite representation. We therefore sample a set of locations $\{\mathbf{r}_i\}_{i=1}^M$, evaluate the fields $(\rho(\mathbf{r}_i), \mathbf{h}(\mathbf{r}_i))$ at those points, and feed the resulting tuples directly into neural networks. This differs from neural

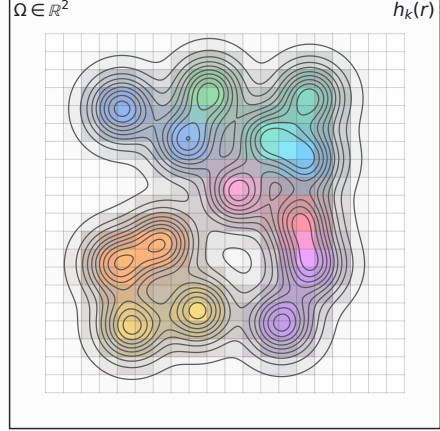

Figure 2: Sampling strategies for evaluating fields. **Left:** Importance sampling draws coordinates in proportion to the density $\rho$, concentrating samples where signal is present. **Right:** Uniform sampling evaluates fields on a fixed grid, covering the domain evenly. In both panels, the curves are isocontours of $\rho(\mathbf{r})$, and the colors show the values of the feature fields $\mathbf{h}_k(\mathbf{r})$.

fields in the usual sense, where signals are encoded implicitly inside a network; here, we work explicitly with sampled field values.

Two sampling approaches are possible: uniform sampling or importance sampling (Figure 2). For molecules in 3D, uniform grids are inefficient: the signal occupies only a small region of space, resolution grows cubically with grid size, and fixed boxes impose artificial boundaries. Instead, we adopt *importance sampling*, drawing locations $\mathbf{r}_i$ with probability proportional to the density field and then evaluating the fields at those points. This concentrates samples where information is present, avoids the need for bounding boxes, and allows the model to learn coordinates and fields jointly for arbitrarily sized molecules.

For regular domains such as images or time series, however, uniform sampling remains natural: pixels form a 2D grid in images, and evenly spaced points define the 1D domain of a signal. In all cases, the model ultimately receives the sampled tuples $\{(\mathbf{r}_i, \rho_i, \mathbf{h}_i)\}_{i=1}^M$.

**Neural architectures for field processing.** We align the choice of neural architecture with the sampling scheme. For molecules, we discretize fields using importance sampling, which produces large unordered point sets. In this setting, we use the Erwin architecture (Zhdanov et al., 2025), a hierarchical, permutation-invariant transformer that scales to thousands of points while preserving global context. For images and time series, inputs lie on regular grids; we use standard 2D and 1D CNNs that exploit locality and run efficiently on uniform samples. This division lets CORDS tackle irregular 3D geometry with a point-based transformer, and lean on compact CNNs where grid structure is natural.

## 4 EXPERIMENTS

We apply CORDS in four settings where variable cardinality naturally arises: molecular generation (QM9 and GeomDrugs), object detection in images (MultiMNIST with out-of-distribution counts), simulation-based inference in astronomy (burst decomposition of light curves), and a synthetic benchmark for local maxima. These tasks span pixel grids for images, three-dimensional space for molecules, time series for light curves, and abstract continuous domains for mathematical functions, all within the same field-based representation. Results for QM9 property regression and for the local-maxima benchmark are deferred to Appendices D.2 and D.1, respectively; additional illustrations and domain-specific details appear in Appendix C. In all experiments, we encode objects

| Model | QM9 | | | | GeomDrugs | |
|---|---|---|---|---|---|---|
| | Atom (%) | Mol (%) | Valid (%) | Unique (%) | Atom (%) | Valid (%) |
| ENF | 85.0 | 4.9 | 40.2 | 98.0 | – | – |
| G-Schnet | 95.7 | 68.1 | 85.5 | 93.9 | – | – |
| GDM | 97.0 | 63.2 | – | – | 75.0 | 90.8 |
| GDM-AUG | 97.6 | 71.6 | 90.4 | 99.0 | 77.7 | 91.8 |
| EDM | 98.7 | 82.0 | 91.9 | 98.7 | 81.3 | 92.6 |
| EDM-Bridge | 98.8 | 84.6 | 92.0 | 98.6 | 82.4 | 92.8 |
| GLDM | 97.2 | 70.5 | 83.6 | 98.9 | 76.2 | 97.2 |
| GLDM-AUG | 97.9 | 78.7 | 90.5 | 98.9 | 79.6 | 98.0 |
| GeoLDM | 98.9 | 89.4 | 93.8 | 98.8 | 84.4 | 99.3 |
| PONITA | 98.9 | 87.8 | – | – | – | – |
| Rapidash | 99.4 | 92.9 | 98.1 | 97.2 | – | – |
| **CORDS** | 97.9 | 82.3 | 91.0 | 97.1 | 78.4 | 94.6 |

Table 1: QM9 and GeomDrugs unconditional generation results, evaluated by the standard RDKit evaluation. Higher is better. Baseline results are adapted from (Xu et al., 2023; Vadgama et al., 2025).

with a Gaussian kernel

$$K(\mathbf{r}; \mathbf{r}_i) \;=\; \exp\left(-\frac{\|\mathbf{r} - \mathbf{r}_i\|^2}{2\sigma^2}\right).$$

### 4.1 MOLECULAR TASKS

**Datasets.** Two benchmarks are considered. QM9 (Ramakrishnan et al., 2014) contains small organic molecules (up to $N{=}29$ heavy atoms) with DFT-computed molecular properties; it is used for both regression and generation tasks. GeomDrugs comprises larger, drug-like molecules covering a broader chemical space and larger atom counts; we use it for unconditional generation to assess scalability and robustness of the proposed framework at higher cardinalities and a setting where modeling additional features, such as charges, is crucial.

**Converting molecules to fields, sampling, and backbone.** Atoms, described by their coordinates and type/charge features, are mapped to density and feature fields $\rho(\mathbf{r})$ and $\mathbf{h}(\mathbf{r})$ using Eq. equation 1. At each training iteration, we discretize these fields by sampling $M$ spatial locations, yielding an unordered set of sampled points

$$\left\{ \left(\mathbf{r}_i,\; \rho_{\mathrm{n}}(\mathbf{r}_i),\; \mathbf{h}_{\mathrm{n}}(\mathbf{r}_i)\right) \right\}_{i=1}^{M},$$

which forms the input to the model (Erwin for all tasks). All learning and sampling steps are performed purely in the field domain, with discrete graphs recovered via decoding only for evaluation metrics that explicitly require molecular graphs. Further details on sampling policies, normalization strategies, hyperparameters, and training schedules are provided in Appendix C.2.

**Unconditional generation.** In the generative setting, both fields and the sampling locations must be modeled explicitly. To achieve this, CORDS learns a joint distribution over coordinates and field values, denoising the entire set $\{(\mathbf{r}_i, \rho_i, \mathbf{h}_i)\}_{i=1}^{M}$. After generation, fields are decoded to molecular graphs using the decoding procedure from the CORDS section: the number of nodes $N$ is estimated from the density mass, node positions are recovered by kernel center fitting, and features are reconstructed via linear projection.

We compare against two distinct groups of baselines, presented separately due to differences in evaluation procedures. Table in Figure 3 compares CORDS to approaches that operate directly on continuous or voxelized representations, such as VoxMol (Pinheiro et al., 2024) and Func-Mol (Kirchmeyer et al., 2025), following their standard sanitization and post-processing evaluation steps. Table 1 provides results according to the standard evaluation criteria common in recent discrete and continuous generative modeling literature (e.g., EDM; Hoogeboom et al. (2022), Rapidash; Vadgama et al. (2025)), where validity, uniqueness, atom-level stability, and molecule-level stability metrics are reported.

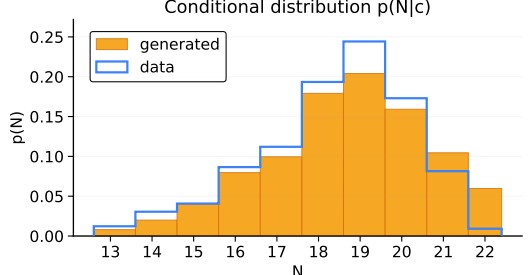

| Model | Atom | Mol | Valid | Unique |
|---|---|---|---|---|
| data | 99.8 | 98.7 | 98.9 | 99.9 |
| VoxMol | 99.2 | 89.3 | 98.7 | 92.1 |
| FuncMol$_{dec}$ | 99.2 | 88.6 | 100.0 | 81.1 |
| FuncMol | 99.0 | 89.2 | 100.0 | 92.8 |
| **CORDS** | 99.2 | 93.8 | 98.7 | 97.1 |

Figure 3: **Left**: Conditional generation on QM9. Histogram of the predicted atom count distribution $p(N|c)$ when conditioning on property ranges unseen during training. **Right**: Unconditional generation results on QM9, evaluated using OpenBabel postprocessing, following VoxMol. Baseline results are adapted from (Kirchmeyer et al., 2025).

| Model | AP | | | AP50 | | | AP75 | | |
|---|---|---|---|---|---|---|---|---|---|
| | In-dist | OOD | Drop (%) | In-dist | OOD | Drop (%) | In-dist | OOD | Drop (%) |
| DETR | 81.2 | 65.4 | 19.5 | 84.0 | 71.7 | 14.6 | 74.2 | 55.1 | 25.8 |
| YOLO | 71.9 | 54.3 | 24.5 | 78.8 | 64.2 | 18.5 | 59.9 | 43.1 | 28.0 |
| **CORDS** | 76.8 | 64.2 | 16.4 | 81.5 | 71.8 | 11.9 | 68.0 | 53.7 | 21.0 |

Table 2: MULTIMNIST object detection results in-distribution vs. OOD. Drop (%) is relative performance decrease.

Finally, we evaluate generalization to larger molecules on GeomDrugs. Here, modeling *non-categorical* atom features, specifically charges, is essential: omitting them harms standard metrics such as validity and atom stability. A strength of CORDS is that continuous features are represented directly in field space and decoded back to graphs, which enables evaluation without postprocessing. In contrast, prior continuous approaches (e.g., VoxMol, FuncMol) typically operate with one-hot atom types and resort to heuristics for charges, which limits comparability on this benchmark; we therefore follow the usual GeomDrugs evaluation protocol. The generative setup mirrors QM9: training and sampling are done entirely in field space, with decoding only for evaluation.

On QM9, CORDS matches or improves upon continuous/voxel baselines (VoxMol, FuncMol) and reaches the overall performance range of E(3)-equivariant GNNs (EDM, GeoLDM, Ponita), despite using a non-equivariant, domain-agnostic backbone. This is the sense in which we describe the results as competitive.

**Conditional generation (QM9).** Most conditional generators are trained by conditioning on target property values and are then evaluated by predicting the properties of generated samples with independent predictors, reporting MAE against the targets (Hoogeboom et al., 2022). When conditioning on a property $c$, one must also model the conditional size distribution $p(N|c)$. Prior work typically discretizes $c$ into bins and treats $(N, c)$ as a joint categorical variable over the number of nodes and the bin of the conditioning variable. This creates a support gap: if a bin is unseen during training, sampling $N$ at that $c$ becomes impossible.

Our approach conditions directly on continuous properties (here, polarizability $\alpha$) without discretizing either $c$ or $N$. Cardinality emerges from field mass, so $p(N|c)$ is learned as part of the conditional field distribution. To test generalization, we remove a range of $c$ during training and condition on that range at inference. Despite the holdout, we recover coherent conditional distribution over the number of atoms, as reflected in the induced atom-count histograms in Fig. 3a.

### 4.2 OBJECT DETECTION (MULTIMNIST)

**Setup.** We demonstrate CORDS on images where the discrete objects of interest are bounding boxes. Each bounding box instance is specified by its centre $(x, y) \in \mathbb{R}^2$ and carries two types of

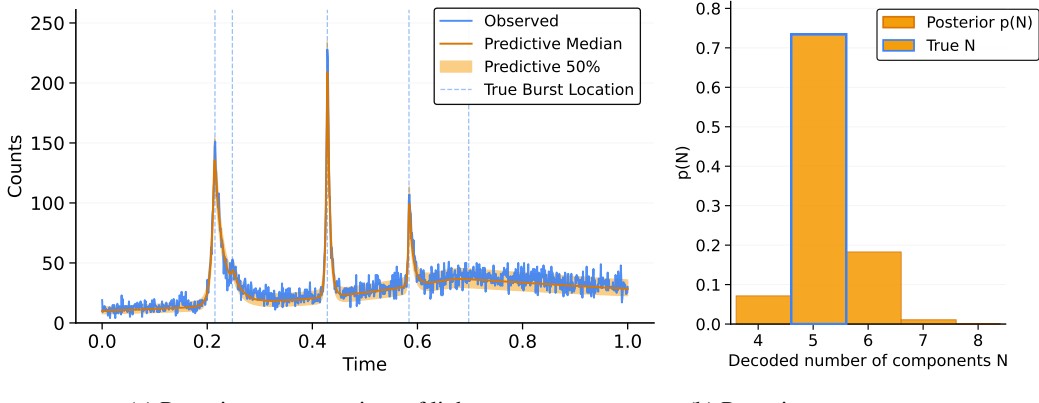

(a) Posterior reconstructions of light curve          (b) Posterior over component count

Figure 4: Simulation-based inference on light curves. **(a)** Observed light curve $\ell$ (blue) with reconstructions from posterior samples $\theta \sim p(\theta \mid \ell)$. Each reconstruction is obtained by decoding sampled fields into component parameters $\theta$ and simulating the resulting light curve. **(b)** Posterior over the number of components $p(N \mid \ell)$.

features: a class label (0–9) and a box shape $(w, h)$. We encode such sets into aligned fields on the image plane: a density field $\rho(\mathbf{r})$, per-class channels that carry one-hot information, and two size channels that store $(w, h)$ where mass is present (Fig. 1). Discrete predictions are recovered with the decoding equations from Section 3. Data is generated on the fly using an online MULTIMNIST generator, avoiding any additional augmentations and effectively yielding an infinite-data regime. Each image contains up to $N_{\max}$ digits (here $N_{\max}{=}15$); digits are uniformly sampled per image, randomly rotated and rescaled, and placed on a black canvas. Ground-truth bounding boxes and classes are the targets for all experiments.

**OOD evaluation.** Beyond standard in-distribution evaluation (images with at most $N_{\max}$ objects), we also construct an out-of-distribution split where the number of digits exceeds the training range. This tests whether a detector can handle variable cardinality without relying on pre-set capacity or a dedicated counting head. Query-based models implicitly cap predictions via their slot budget; once the scene contains more objects than slots, additional instances are necessarily missed. In CORDS, the cardinality is encoded by density mass, so increasing scene density is naturally reflected in the representation, and the decoding remains valid for larger scenes without changing the network.

**Training objective.** We train by minimizing a pixel-wise mean squared error (MSE) on both the density and feature fields, combined with a penalty on mismatched counts. The overall loss is

$$\mathcal{L} = \mathcal{L}_{\mathrm{MSE}} + \lambda \, (\hat{N} - N)^2,$$

where the predicted count $\hat{N}$ is given by the total mass of the density field,

$$\hat{N} = \int \rho(x, y) \, dx \, dy.$$

This way, the number of objects is treated as a continuous, differentiable quantity and optimized jointly with the other objectives.

**Baselines and metrics.** We compare to a DETR detector with a fixed query budget and to a compact anchor-free YOLO variant. All methods are competitive in-distribution. Under OOD counts, the fixed-query baseline underestimates due to capacity limits, while CORDS continues to track the true cardinality more accurately. We report the standard detection metrics in Table 2. For fair comparison, we allocated all networks with a total of 8 million parameters. The exact implementation details of the baselines are discussed in Appendix C.3.

### 4.3 SIMULATION-BASED INFERENCE FOR FRBs

Simulation-based inference (SBI) deals with settings where the likelihood is unavailable but simulation from the generative process is possible: we draw parameters from a prior, generate observations $\ell$, and train a conditional model to approximate the posterior $p(\theta|\ell)$. We adopt flow matching for posterior estimation (Dax et al., 2023), which learns a time-dependent vector field that transports a simple base distribution to the target posterior, giving us amortized inference with a tractable density.

We demonstrate this approach on the problem of modeling Fast Radio Bursts (FRBs). FRBs are short, millisecond-scale flashes of radio emission of extragalactic origin, whose astrophysical mechanisms are still not fully understood Petroff et al. (2022); Zhang (2023). They are typically modeled as a superposition of a variable number of transient components, each characterized by parameters such as onset time, amplitude, rise time, and skewness. Recovering the posterior over these parameters given noisy photon-count light curves is a natural setting for SBI.

In our experiment we work with 1D photon-count light curves with Poisson noise. We first sample the number of burst components $N$ from a uniform prior. Given $N$, each component has parameters $\theta = (t_0, A, \tau_{\text{rise}}, \text{skew})$ drawn from astrophysical priors. If $N$ were fixed, this problem would be straightforward: for example, we could represent each component as a token of dimension $\mathbb{R}^4$ and train a transformer with flow matching conditioned on $\ell$. The challenge is that $N$ varies, so we also need to recover $p(N \mid \ell)$, which is not trivial. This is the motivation for our approach.

**Training and inference.** We map bursts into continuous fields on the time axis: a density $\rho(t)$ with peaks at onset times and a feature field $\mathbf{h}(t)$ carrying $(A, \tau_{\text{rise}}, \text{skew})$ on the same support. Each light curve is discretized on a uniform grid of $K$ points ($K=1000$), where we evaluate $(\rho, \mathbf{h})$ and append the observed counts $\ell(t)$. We then train a flow-matching model to approximate $p(\rho(t), \mathbf{h}(t) \mid \ell)$ directly in field space; at inference we sample fields and decode them into component sets. Appendix C.4 details the encoding and decoding. Figure 4 shows posterior light-curve reconstructions and the induced distribution $p(N \mid \ell)$.

## 5 DISCUSSION

**Implications and takeaways.** Our experiments show that field-based learning handles variable cardinality reliably across domains. In molecular *generation*, CORDS attains competitive results against well-established GNN baselines on QM9 and GeomDrugs, while outperforming prior continuous approaches such as VoxMol on QM9. A key advantage is that non-categorical atomic features, such as partial charges, can be modeled directly in the feature fields and decoded back. In *object detection*, CORDS exhibits a smaller performance drop under out-of-distribution object counts, plausibly because cardinality is encoded as density mass and can be regularized with a simple count penalty, making the representation more stable as scenes become denser. For *simulation-based inference* on light curves, the approach sidesteps explicit modeling of $p(N|\ell)$: the posterior over cardinalities arises naturally from the learned field distribution, simplifying training and inference.

**Limitations.** CORDS incurs practical costs. High-fidelity reconstruction in molecules benefits from dense sampling ($\sim 10^3$ points per molecule), which makes direct scaling to larger graphs computationally expensive. Accuracy also depends on precise kernel-center localization; refinements (e.g., L-BFGS) help but add latency, creating a speed–accuracy trade-off. In detection, overlapping kernels can hinder separation of nearby objects, requiring fine-tuning kernel widths. We further discuss and explain these limitations in Appendix C.2.

**Future work.** Several directions follow naturally. On the *detection* side, evaluating CORDS on larger-scale benchmarks (e.g., COCO) would test robustness under heavy occlusion, class diversity, and crowding, and could explore learned, spatially adaptive kernels to separate nearby instances. For *molecular* modeling, extending conditional generation to richer tasks, such as pocket-conditioned ligand design, regional inpainting, or multi-property control would further probe the benefits of working in field space with continuous attributes (charges, spins, partial occupancies). More broadly, we expect CORDS to be most useful in *hybrid* settings that combine discrete and continuous modalities, for instance, molecules or materials described as discrete atomic graphs alongside continuous

DFT-derived electron densities (Elsborg et al., 2026), since CORDS provides a way to map the discrete structure into a continuous field representation that naturally aligns with the density modality.

## 6 CONCLUSION

We introduced CORDS, a framework for modeling variable-size sets through continuous fields, where positions and features are mapped onto density and feature fields and cardinality is recovered directly from total density mass. The appeal lies in its simplicity: a single field-based representation captures counts, locations, and attributes, enabling learning entirely in field space while still allowing exact recovery of discrete predictions across molecules, images, and simulated astronomy data.

Compared to earlier continuous-representation methods in computer vision and molecular modeling, CORDS achieves competitive performance while offering the flexibility to represent arbitrary features beyond predefined types. Overall, the results point to a single field-based representation as an elegant and broadly applicable alternative: the same encoding (density and feature fields), decoding (mass for counts, kernel centers for positions, projections for attributes), and training objectives carry across tasks, providing a unified solution to modeling variable cardinality.

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

# A  CORDS THEORETICAL FRAMEWORK

We collect precise assumptions, definitions, and proofs validating the decoding steps and the duality between finite sets and continuous fields. To aid readability, each formal statement is followed by a brief intuition.

**General intuition.** A finite set of elements with positions and features is encoded as a pair of continuous fields by superimposing a fixed kernel at each position: a scalar density $\rho$ and a feature field $\mathbf{h}$ that uses the *same* kernels. Decoding proceeds in three deterministic steps: (i) read off the cardinality from the total mass of $\rho$, (ii) recover the positions by matching $\rho$ with a sum of kernel translates, and (iii) recover the features by projecting $\mathbf{h}$ onto the recovered kernel translates, which yields a small linear system with Gram matrix $G$. Uniqueness comes from standard identifiability of equal-weight kernel mixtures and positive-definiteness of $G$.

## A.1  CORDS DECODING PROOFS

### A.1.1  RECOVERING THE NUMBER OF ELEMENTS

**Proposition A.1** (Cardinality). Let $S \in \mathcal{S}_\Omega$ and $\Phi(S) = (\rho, \mathbf{h})$. Then $\int_\Omega \rho(\mathbf{r}) \, d\mathbf{r} = |S|$.

*Proof.* By linearity of the integral and the definition of $\alpha$,

$$\int_\Omega \rho(\mathbf{r}) \, d\mathbf{r} = \frac{1}{\alpha} \sum_{i=1}^N \int_\Omega K(\mathbf{r}; \mathbf{r}_i) \, d\mathbf{r} = \frac{1}{\alpha} \sum_{i=1}^N \alpha = N.$$

$\square$

**Intuition.** Every element contributes to one kernel "bump" whose total mass is $\alpha$. Adding them and dividing by $\alpha$ gives the count.

### A.1.2  POSITIVE–DEFINITENESS OF THE GRAM MATRIX

**Lemma A.2** (Gram matrix is SPD). For distinct centers $\{\mathbf{r}_i\}_{i=1}^N$ and $\kappa_i(\cdot) = K(\cdot; \mathbf{r}_i)$, the matrix $G_{ij} = \int_\Omega \kappa_i(\mathbf{r}) \, \kappa_j(\mathbf{r}) \, d\mathbf{r}$ is symmetric positive–definite.

*Proof.* Symmetry is immediate. For any $\mathbf{c} \in \mathbb{R}^N \setminus \{0\}$, let $\phi_{\mathbf{c}}(\mathbf{r}) = \sum_i c_i \kappa_i(\mathbf{r})$. Then

$$\mathbf{c}^\top G \mathbf{c} = \int_\Omega \phi_{\mathbf{c}}(\mathbf{r})^2 \, d\mathbf{r} \ \geq \ 0.$$

If $\mathbf{c}^\top G \mathbf{c} = 0$, then $\phi_{\mathbf{c}} = 0$ in $L^2$, hence a.e.; by (A2) the translates are linearly independent, so $\mathbf{c} = 0$, a contradiction. Thus $G$ is SPD. $\square$

**Intuition.** Think of $\{\kappa_i\}$ as a set of directions in a Hilbert space. Their Gram matrix computes inner products. If no nontrivial combination of the $\kappa_i$ cancels out, the quadratic form $\mathbf{c}^\top G \mathbf{c}$ is strictly positive for all nonzero $\mathbf{c}$.

### A.1.3  EXACT RECOVERY OF FEATURES

**Proposition A.3** (Feature inversion with correct $\alpha$). Let $S \in \mathcal{S}_\Omega$ with $\Phi(S) = (\rho, \mathbf{h})$ and distinct centers $\{\mathbf{r}_i\}$. Define $G, B$ from the recovered centers as above. Then

$$B = \frac{1}{\alpha} G \mathbf{X} \qquad \text{and} \qquad \mathbf{X} = \alpha \, G^{-1} B,$$

so the recovered features equal the original features.

*Proof.* From $\mathbf{h}(\mathbf{r}) = \frac{1}{\alpha} \sum_j \mathbf{x}_j \, \kappa_j(\mathbf{r})$,

$$B_{i:} = \int_\Omega \mathbf{h}(\mathbf{r}) \, \kappa_i(\mathbf{r}) \, d\mathbf{r} = \frac{1}{\alpha} \sum_j \mathbf{x}_j \int_\Omega \kappa_j(\mathbf{r}) \, \kappa_i(\mathbf{r}) \, d\mathbf{r} = \frac{1}{\alpha} \sum_j G_{ij} \, \mathbf{x}_j.$$

Stacking rows gives $B = \frac{1}{\alpha} G \, \mathbf{X}$. Invertibility follows from Lemma A.2, yielding $\mathbf{X} = \alpha G^{-1} B$. $\quad\square$

**Intuition.** Because $\mathbf{h}$ is a linear combination of the same kernels that make up $\rho$, projecting $\mathbf{h}$ onto each kernel recovers the corresponding coefficient vector. The Gram matrix accounts for overlap between kernels; its inverse untangles that overlap.

### A.1.4 RECOVERING POSITIONS FROM THE DENSITY

**Proposition A.4** (Position recovery). Let $S \in \mathcal{S}_\Omega$ with $\Phi(S) = (\rho, \mathbf{h})$ and $N = |S|$. If (A3) holds, then the minimizers of equation 3 are exactly the ground-truth centers up to permutation, and the optimal value is 0.

*Proof.* For the ground-truth centers $\{\mathbf{r}_i\}$, the integrand in equation 3 vanishes pointwise by construction, so the objective equals 0. Conversely, any minimizer with value 0 satisfies $\rho(\cdot) = \frac{1}{\alpha} \sum_{i=1}^N K(\cdot; \mathbf{r}_i^\star)$, hence $\sum_i K(\cdot; \mathbf{r}_i) = \sum_i K(\cdot; \mathbf{r}_i^\star)$. By (A3) the sets of centers coincide up to permutation. $\quad\square$

**Intuition.** The density $\rho$ must be explainable as a sum of identical shapes (kernels) placed somewhere in $\Omega$. If equal-weight mixtures are unique, there is only one way to place $N$ such shapes to obtain exactly $\rho$: at the original centers (order irrelevant).

## B  CORDS FRAMEWORK FOR GRAPHS

In order to extend our work to graphs, or data with discrete relational objects (such as edges), we will define **Field of Graph** as a quadruple

$$f = \big(\rho_\text{n}, \rho_\text{e}, \mathbf{h}_\text{n}, \mathbf{h}_\text{e}\big)$$

where

- $\rho_\text{n} : \Omega \to \mathbb{R}_{\geq 0}$ is the *node density*;
- $\rho_\text{e} : \Omega \times \Omega \to \mathbb{R}_{\geq 0}$ is the *edge density*;
- $\mathbf{h}_\text{n} : \Omega \to \mathbb{R}^{d_x}$ is a vector-valued *node feature field*.
- $\mathbf{h}_\text{e} : \Omega \times \Omega \to \mathbb{R}^{d_y}$ is a vector-valued *edge feature field*.

We now specify a concrete construction of the encoding map $\Phi : \mathcal{G}_\Omega \to \mathcal{F}$. This construction associates to each graph a distributional representation over the domain $\Omega$, using fixed spatial kernels to define the node density, edge density, and feature fields.

Let us fix two continuous, positive kernels:

$$k_\text{n} : \Omega \times \Omega \; \to \; \mathbb{R}_{\geq 0}, \qquad\qquad k_\text{e} : (\Omega \times \Omega)^2 \; \to \; \mathbb{R}_{\geq 0}.$$

The kernel $k_\text{n}$ determines how node mass is distributed across space, while $k_\text{e}$ governs the representation of edges.

Given a geometric graph $G = (V, E, \boldsymbol{X}, \boldsymbol{Y}, \boldsymbol{R}) \in \mathcal{G}_\Omega$, we define its field representation

$$\Phi(G) = \big(\rho_\text{n}, \rho_\text{e}, \mathbf{h}_\text{n}, \mathbf{h}_\text{e}\big)$$

as

$$\rho_\text{n}(\boldsymbol{r}) = \sum_{v \in V} k_\text{n}(\boldsymbol{r}; \boldsymbol{r}_v), \qquad \rho_\text{e}(\boldsymbol{r}_1, \boldsymbol{r}_2) = \sum_{(u,v) \in E} k_\text{e}\big((\boldsymbol{r}_1, \boldsymbol{r}_2); (\boldsymbol{r}_u, \boldsymbol{r}_v)\big),$$

$$\mathbf{h}_\text{n}(\boldsymbol{r}) = \sum_{v \in V} \mathbf{x}_v k_\text{n}(\boldsymbol{r}; \boldsymbol{r}_v), \qquad \mathbf{h}_\text{e}(\boldsymbol{r}_1, \boldsymbol{r}_2) = \sum_{(u,v) \in E} \mathbf{y}_{uv} k_\text{e}\big((\boldsymbol{r}_1, \boldsymbol{r}_2); (\boldsymbol{r}_u, \boldsymbol{r}_v)\big). \tag{5}$$

**Special case: Dirac kernels.**   Choosing

$$k_{\mathrm{n}}(\boldsymbol{r}; \boldsymbol{r}_v) = \delta(\boldsymbol{r} - \boldsymbol{r}_v), \qquad k_{\mathrm{e}}((\boldsymbol{r}_1, \boldsymbol{r}_2); (\boldsymbol{r}_u, \boldsymbol{r}_v)) = \delta(\boldsymbol{r}_1 - \boldsymbol{r}_u)\,\delta(\boldsymbol{r}_2 - \boldsymbol{r}_v),$$

recovers the standard discrete graph structure in distributional form. This limiting case shows that our construction generalizes the original discrete graph while embedding it in a continuous domain. With a suitable choice of interactions, we obtain the traditional message passing.

## B.1   Continuous Graph Convolution and the Discrete Message–Passing Limit

Throughout this appendix we work on an ambient domain $\Omega \subseteq \mathbb{R}^d$. A geometric graph $G = (V, E, \mathbf{R}, \mathbf{X})$ with node positions $\mathbf{r}_v \in \Omega$ and node features $\mathbf{h}_v^{(k)} \in \mathbb{R}^{d_h}$ is encoded into continuous objects

$$\rho_{\mathrm{n}},\ \rho_{\mathrm{e}},\ \mathbf{h}^{(k)} : \Omega \longrightarrow \mathbb{R}^{d_h}$$

In particular, for a *node kernel* $k_{\mathrm{n}}$ and an *edge kernel* $k_{\mathrm{e}}$ we have

$$\begin{aligned}
\rho_{\mathrm{n}}(\mathbf{r}) &= \sum_{u \in V} k_{\mathrm{n}}(\mathbf{r}; \mathbf{r}_u), \\
\rho_{\mathrm{e}}(\mathbf{r}_1, \mathbf{r}_2) &= \sum_{(u,v) \in E} k_{\mathrm{e}}((\mathbf{r}_1, \mathbf{r}_2); (\mathbf{r}_u, \mathbf{r}_v)), \\
\mathbf{h}^{(k)}(\mathbf{r}) &= \sum_{u \in V} \mathbf{h}_u^{(k)}\, k_{\mathrm{n}}(\mathbf{r}; \mathbf{r}_u).
\end{aligned} \tag{6}$$

### B.1.1   Continuous convolution

Given a field feature $\mathbf{h}^{(k)} : \Omega \to \mathbb{R}^{d_h}$ at layer $k$, the *continuous graph convolution* introduced in

$$\mathbf{h}^{(k+1)}(\mathbf{r}_i) \;=\; \sigma\Big(\mathbf{W} \underbrace{\int_{\Omega} \mathbf{h}^{(k)}(\mathbf{r}_j)\, \rho_{\mathrm{e}}(\mathbf{r}_i, \mathbf{r}_j)\, \mathrm{d}\mathbf{r}_j}_{=:\ \mathbf{m}^{(k)}(\mathbf{r}_i)}\Big), \tag{7}$$

where $\mathbf{W} \in \mathbb{R}^{d_h \times d_h}$ is a trainable linear map, $\sigma$ is any point-wise non-linearity (ReLU, SiLU, …), and $\mathbf{m}^{(k)}$ denotes the *message field* aggregated from all spatial locations.

### B.1.2   Dirac kernels and the discrete limit

We now take

$$k_{\mathrm{n}}(\mathbf{r}; \mathbf{r}_u) \;=\; \delta(\mathbf{r} - \mathbf{r}_u), \qquad k_{\mathrm{e}}((\mathbf{r}_1, \mathbf{r}_2); (\mathbf{r}_u, \mathbf{r}_v)) = \delta(\mathbf{r}_1 - \mathbf{r}_u)\,\delta(\mathbf{r}_2 - \mathbf{r}_v),$$

i.e. each node (edge) is represented by a Dirac delta of unit mass centred at its position.

**Proposition B.1** (Continuous convolution $\longrightarrow$ message passing). *Let $k_{\mathrm{n}}, k_{\mathrm{e}}$ be the Dirac kernels above. Then, evaluating equation 7 at the node positions $\mathbf{r}_i = \mathbf{r}_u$ yields*

$$\mathbf{h}_u^{(k+1)} \;=\; \sigma\Big(\mathbf{W} \underbrace{\sum_{v \in V} e_{uv}\, \mathbf{h}_v^{(k)}}_{=:\ \mathbf{m}_u^{(k)}}\Big), \tag{8}$$

*where $e_{uv} = 1$ if $(u, v) \in E$ and $0$ otherwise. That is, the continuous convolution reduces exactly to the standard message-passing update with sum aggregation.*

*Proof.* Using the encoding equation 6 with Dirac kernels,

$$\rho_{\mathrm{e}}(\mathbf{r}_i, \mathbf{r}_j) \;=\; \sum_{(p,q) \in E} \delta(\mathbf{r}_i - \mathbf{r}_p)\,\delta(\mathbf{r}_j - \mathbf{r}_q).$$

Fix a node $u$ and set $\mathbf{r}_i = \mathbf{r}_u$. Substituting into the message integral in equation 7 gives

$$\mathbf{m}^{(k)}(\mathbf{r}_u) = \int_\Omega \mathbf{h}^{(k)}(\mathbf{r}_j) \sum_{(p,q)\in E} \delta(\mathbf{r}_u - \mathbf{r}_p)\,\delta(\mathbf{r}_j - \mathbf{r}_q)\,\mathrm{d}\mathbf{r}_j$$

$$= \sum_{(p,q)\in E} \delta(\mathbf{r}_u - \mathbf{r}_p)\,\mathbf{h}^{(k)}(\mathbf{r}_q)$$

$$= \sum_{(p,q)\in E} \delta_{up}\,\mathbf{h}_q^{(k)} \;=\; \sum_{v\in V} e_{uv}\,\mathbf{h}_v^{(k)},$$

where $\delta_{up}$ is the Kronecker delta (the Dirac delta evaluates to 1 iff $\mathbf{r}_u = \mathbf{r}_p$, or equivalently $u = p$). Finally, plugging $\mathbf{m}^{(k)}(\mathbf{r}_u) = \mathbf{m}_u^{(k)}$ back into equation 7 gives equation 8. $\qquad\square$

**Remark 1** (Vanishing-width Gaussian kernels)**.** If $k_\mathrm{n}$ and $k_\mathrm{e}$ are isotropic Gaussians of width $\sigma$ (as used in Eq. (9) of the main text), then $k_\mathrm{n}, k_\mathrm{e} \xrightarrow{\sigma\to 0}$ Dirac distributions in the sense of tempered distributions. Therefore the continuous convolution converges to the message-passing update equation 8 as $\sigma \to 0$.

## B.2 EXTENDING CORDS TO NON-GEOMETRIC GRAPHS

The core construction of CORDS relies on the existence of an explicit geometric embedding $p :$ $V \to \Omega$, which maps each node to a position in a continuous domain. However, many real-world graphs do not come equipped with natural spatial coordinates. To extend our framework to such non-geometric graphs, we propose using spectral embeddings derived from the graph's topology.

### B.2.1 SPECTRAL EMBEDDINGS VIA GRAPH LAPLACIAN

Given a graph $G = (V, E, \boldsymbol{X})$ without predefined node positions, we compute a spectral embedding based on the graph Laplacian. Specifically, let $A \in \mathbb{R}^{|V|\times|V|}$ be the adjacency matrix and $D$ the diagonal degree matrix with $D_{ii} = \deg(i)$. The (unnormalized) graph Laplacian is defined as:

$$L = D - A.$$

Let $\{\lambda_i\}_{i=1}^{|V|}$ be the eigenvalues of $L$, with corresponding eigenvectors $\{\boldsymbol{v}_i\}_{i=1}^{|V|}$. We define a spectral embedding

$$p_{\mathrm{spec}}(v) = \big(\boldsymbol{v}_2(v), \boldsymbol{v}_3(v), \ldots, \boldsymbol{v}_{d+1}(v)\big),$$

where $\boldsymbol{v}_i(v)$ denotes the $v$-th entry of the $i$-th eigenvector. The first non-trivial eigenvectors capture global structural information, positioning nodes with similar topological roles close to each other in $\mathbb{R}^d$.

This spectral embedding provides a continuous proxy for node positions, enabling us to apply the same CORDS construction as in the geometric case. Effectively, it allows us to treat arbitrary graphs as if they were embedded in a geometric space, lifting them into a continuous field representation.

### B.2.2 LIMITATIONS AND PRACTICAL CONSIDERATIONS

While spectral embeddings offer a principled way to introduce geometry into non-geometric graphs, they come with trade-offs. Specifically:

- The embedding dimensionality $d$ is a design choice. Using fewer dimensions provides a compressed view of the graph's topology, which can be sufficient for downstream tasks like regression or classification.

- However, reducing $d$ also means that the bijective property of the Graph–Field dual pair is lost, as the original graph cannot be perfectly reconstructed from the continuous representation.

- For applications where reversible mapping is not critical (e.g., predictive tasks on node-level or graph-level properties), this trade-off is acceptable. On the other hand, generative tasks that require recovering discrete graph structure from fields would necessitate higher-dimensional embeddings or alternative encoding strategies.

In summary, spectral embeddings allow us to extend the CORDS to general graphs without prede-fined coordinates, enabling continuous representations even in the absence of natural geometry.

## C  ADDITIONAL EXPERIMENTAL DETAILS

### C.1  APPROXIMATING THE INVERSE DECODING IN PRACTICE

The exact inversion formulas in Appendix A involve domain integrals and solving linear systems built from kernel inner products. When fields are only available at finitely many sample locations (either on a grid or from a sampler) these integrals are approximated by Monte Carlo (MC). Below we describe the practical decoding we use in all experiments. It proceeds in three steps: *(i) estimate the number of elements $N$, (ii) recover their positions*, and *(iii) reconstruct their feature vectors*. We emphasise the intuition at each step and show how importance sampling enters the feature inversion.

**Notation and sampling setup.**  Let $\{\,\mathbf{r}_s\,\}_{s=1}^S \subset \Omega$ be the evaluation points (*field samples*). They are drawn either *uniformly* from a bounding box of $\Omega$, or by *importance sampling* from a proposal $q(\mathbf{r})$ proportional to a kernel mixture centred near the (unknown) element locations. We write $\rho(\mathbf{r}_s)$ and $\mathbf{h}(\mathbf{r}_s)$ for the sampled density and feature fields at those points. In all cases, MC weights are taken as

$$w_s \;=\; \frac{1}{S\,q(\mathbf{r}_s)} \quad \text{(importance sampling)}, \qquad w_s \;=\; \frac{1}{S} \quad \text{(uniform over a box)}.$$

*Remark:* for uniform sampling the mathematically unbiased weight is $\frac{\mathrm{Vol}(\Omega_{\mathrm{box}})}{S}$; in our implementation we use $\frac{1}{S}$ and rely on the fact that the unknown constant volume multiplies *both* sides of the linear system in Step (iii) and cancels out in the solve.

### (i) Estimating the number of elements $\widehat{N}$ and normalising $\rho$.

The subsequent position fit needs to know *how many kernels* to place. Hence we estimate $N$ directly from the sampled density before any other step.

In theory, $N = \int_\Omega \rho(\mathbf{r})\, d\mathbf{r}$. With sampling $\mathbf{r}_s \sim q$ the Monte Carlo (MC) estimator is

$$\widehat{N}_{\mathrm{MC}} \;=\; \sum_{s=1}^S \rho(\mathbf{r}_s)\, w_s \;=\; \frac{1}{S} \sum_{s=1}^S \frac{\rho(\mathbf{r}_s)}{q(\mathbf{r}_s)},$$

where $w_s = \frac{1}{Sq(\mathbf{r}_s)}$ for importance sampling and $w_s = \frac{1}{S}$ for uniform sampling over a box (up to an unknown constant volume factor).

For importance sampling, $q$ is typically only known up to a global constant ($q \propto \sum_u \kappa_{\mathrm{prop}}(\cdot\,;\mathbf{r}_u)$); for uniform sampling, the box volume enters $q$. In practice we avoid carrying these constants by working with a *calibrated* density: during training the encoder $\Phi$ optionally rescales $\rho$ so that its sample mean equals the *true* cardinality. At test time we therefore set

$$\widehat{N} \;=\; \mathrm{round}\!\left( \frac{1}{S} \sum_{s=1}^S \rho(\mathbf{r}_s) \right).$$

Because of MC noise $\frac{1}{S}\sum_s \rho(\mathbf{r}_s)$ will rarely be an exact integer; empirically it concentrates within $\pm 0.5$ of the truth, so rounding is appropriate.[1]

*Kernel-quadrature alternative.*  When the density field is (approximately) a superposition of known kernel translates, one can also estimate $\int_\Omega \rho(\mathbf{r})\, d\mathbf{r}$ without explicit knowledge of $q$ by using *kernel quadrature* (a.k.a. Bayesian / GP quadrature). Let $K(\cdot\,;\cdot)$ be the kernel used by the encoder, and assume its mass is (center-)independent,

$$\alpha \;:=\; \int_\Omega K(\mathbf{r};\mathbf{s})\, d\mathbf{r} \quad \text{for all } \mathbf{s} \in \Omega.$$

---

[1] If a calibration pass is not available, one can use the generic $\widehat{N}_{\mathrm{MC}}$ above with explicit $w_s = \frac{1}{Sq(\mathbf{r}_s)}$; for importance sampling, the unknown proportionality constant cancels after the normalisation step below.

Define the kernel matrix on the sample locations $\mathbf{r}_s$ by

$$\mathbf{K}_{st} := K(\mathbf{r}_s; \mathbf{r}_t), \qquad s, t \in \{1, \dots, S\}.$$

We compute quadrature weights $w \in \mathbb{R}^S$ by matching the integrated kernel sections,

$$(\mathbf{K} + \varepsilon \mathbf{I})\, w = \alpha \, \mathbf{1},$$

with a small ridge $\varepsilon > 0$ for numerical stability. The resulting estimator

$$\widehat{N}_{\mathrm{KQ}} = \sum_{s=1}^{S} w_s \, \rho(\mathbf{r}_s)$$

is *exact* whenever $\rho$ lies in the span of $\{K(\cdot; \mathbf{r}_s)\}_{s=1}^{S}$, and provides a principled, kernel-native approximation otherwise. This option removes the need to track proposal normalisation constants while still leveraging the known kernel shape. We use the calibrated-mean estimate in the main experiments for simplicity, and kernel quadrature as an optional alternative when calibration is not available or when sample distributions are highly non-uniform.

**(ii) Recovering positions.**

*Initialisation by a mixture fit.* Given the samples $\{(\mathbf{r}_s, \rho(\mathbf{r}_s))\}_{s=1}^{S}$ and the estimate $\widehat{N}$, we fit an $\widehat{N}$-component *isotropic* Gaussian mixture model (GMM) to the $\mathbf{r}_s$, initialised with $k$-means++ on coordinates. The resulting means $\{\tilde{\mathbf{r}}_u\}_{u=1}^{\widehat{N}}$ are coarse location estimates. (Optionally we search over $\widehat{N} \pm \delta$ components by BIC and pick the best, but we keep $\widehat{N}$ unless BIC strongly prefers a neighbour.)

Here BIC denotes the Bayesian information criterion, $\mathrm{BIC}(k) = -2\log L_k + p_k \log S$, where $L_k$ is the maximized likelihood of a $k$-component GMM with $p_k$ free parameters fitted to $S$ samples.

*Refinement by kernel matching.* We then refine the centres by minimising the squared discrepancy between the observed density and a superposition of kernel translates. Writing $\kappa(\mathbf{r}; \mathbf{r}_u) = K(\mathbf{r}; \mathbf{r}_u)$ and allowing a global amplitude $a > 0$ to absorb small normalisation mismatches (e.g., boundary truncation, unknown $\alpha$), we minimise

$$\mathcal{L}_{\mathrm{pos}}(\{\mathbf{r}_u\}, a) = \frac{1}{S} \sum_{s=1}^{S} \left( \rho(\mathbf{r}_s) - a \frac{1}{\alpha} \sum_{u=1}^{\widehat{N}} \kappa(\mathbf{r}_s; \mathbf{r}_u) \right)^2. \tag{9}$$

We run LBFGS for at most 50 iterations starting from the GMM means. When the dynamic range is large, we minimise the same objective in *log space* (i.e., replace both terms by their $\log$, with a small floor), which stabilises the fit of $a$ and the centres near sharp peaks.

*Intuition.* Step (ii) exactly mirrors the theoretical position recovery (Appendix A): we seek the unique set of centres whose kernel sum reproduces the observed $\rho$. The GMM gives a good basin of attraction; LBFGS then snaps the centres onto the mode locations determined by $\rho$.

**(iii) Reconstructing feature vectors from h.**

*Theory recap.* For the recovered centres $\{\mathbf{r}_u\}_{u=1}^{\widehat{N}}$ define $\kappa_u(\mathbf{r}) = K(\mathbf{r}; \mathbf{r}_u)$. The ideal ($L^2$) projection used in Appendix A reads

$$G_{uv} = \int_{\Omega} \kappa_u(\mathbf{r}) \, \kappa_v(\mathbf{r}) \, d\mathbf{r}, \qquad B_{u:} = \int_{\Omega} \mathbf{h}(\mathbf{r}) \, \kappa_u(\mathbf{r}) \, d\mathbf{r}, \qquad \mathbf{X} = \alpha \, G^{-1} B.$$

In code we approximate both integrals by MC with the *same* weights $w_s$. *Monte Carlo feature inversion.* With samples $\mathbf{r}_s \sim q$ and weights $w_s = \frac{1}{Sq(\mathbf{r}_s)}$, define

$$\widehat{G}_{uv} = \sum_{s=1}^{S} \kappa_u(\mathbf{r}_s) \, \kappa_v(\mathbf{r}_s) \, w_s, \tag{10}$$

$$\widehat{B}_{u:} = \sum_{s=1}^{S} \mathbf{h}(\mathbf{r}_s) \, \kappa_u(\mathbf{r}_s) \, w_s. \tag{11}$$

We then recover the feature matrix by solving the $\widehat{N} \times \widehat{N}$ system

$$\widehat{G}\,\widetilde{\mathbf{X}} \;=\; \frac{1}{\alpha}\,\widehat{B}, \qquad \text{and setting} \qquad \widehat{\mathbf{X}} \;=\; \alpha\,\widetilde{\mathbf{X}}. \tag{12}$$

This expression is valid for any $\alpha > 0$ and reduces to the usual solve $\widehat{G}\,\widehat{\mathbf{X}} = \widehat{B}$ when $\alpha = 1$.

In our implementation we experimented with three standard radial kernels: Gaussian, Laplacian, and Epanechnikov, each normalised to unit mass, so that $\alpha = 1$ and we simply solve $\widehat{G}\,\widehat{\mathbf{X}} = \widehat{B}$. Concretely, these kernels have the usual forms

$$\kappa_{\text{Gauss}}(\mathbf{r}) \propto \exp\!\big(-\|\mathbf{r}\|_2^2/2\sigma^2\big), \qquad \kappa_{\text{Lap}}(\mathbf{r}) \propto \exp\!\big(-\|\mathbf{r}\|_2/\sigma\big), \qquad \kappa_{\text{Epan}}(\mathbf{r}) \propto \max\!\big(0,\, 1-\|\mathbf{r}\|_2^2/\sigma^2\big),$$

where $\sigma$ is a bandwidth parameter. All three are positive, radially symmetric, and compactly or effectively compactly supported, and in our experiments they lead to very similar decoding behaviour; we therefore use Gaussians by default and view Laplacian/Epanechnikov kernels as interchangeable alternatives.

For numerical robustness we add a tiny diagonal $\varepsilon I$ to $\widehat{G}$ ($\varepsilon$ is $10^{-4}$ times the average diagonal) and fall back to least squares if a direct solve fails. When element types are categorical (e.g., one-hot), we take $\mathrm{argmax}$ over the feature channels of each row of $\widehat{\mathbf{X}}$.

*Proposal normalization.* If $q$ is known only up to a constant (importance sampling) or includes an unknown box volume (uniform), $w_s$ is known up to the same constant factor $c$. Both $\widehat{G}$ and $\widehat{B}$ in equation 10–equation 11 are multiplied by $c$, which cancels in the linear system $\widehat{G}\,\widehat{\mathbf{X}} = \widehat{B}$. This is why using $w_s = \frac{1}{S}$ under uniform sampling is sufficient in practice, and why we can implement importance weights with an unnormalised mixture $q \propto \sum_u \kappa_{\text{prop}}(\cdot; \mathbf{r}_u)$.

**Summary.**

1. *Cardinality:* estimate $\widehat{N}$ from the sample mean of $\rho$, round to the nearest integer (non-integral values within $\pm 0.5$ are expected).

2. *Positions:* fit an $\widehat{N}$-component isotropic GMM to the sample coordinates and refine the means by minimising equation 9 with LBFGS (optionally in log-space, with a global amplitude $a$).

3. *Features:* form $\widehat{G}, \widehat{B}$ by the MC formulas equation 10–equation 11 using the same weights $w_s$ as for the integral, solve $\widehat{G}\,\widehat{\mathbf{X}} = \widehat{B}$ (or $\alpha\,\widehat{G}^{-1}\widehat{B}$ if non-unit kernels are used), and post-process categorical channels by $\mathrm{argmax}$.

**Practical notes (hyperparameters).** We initialise the GMM by $k$-means++, search over $\widehat{N} \pm \delta$ components with $\delta \approx 0.15\,\widehat{N}$ unless $\widehat{N}$ is very small, and run LBFGS for at most 50 iterations with a Wolfe line search. Importance sampling uses the same kernel family as the density with a proposal bandwidth (`sample_sigma`) close to the encoding bandwidth; temperature-sharpening of the proposal probabilities helps focus samples near density peaks; overall complexity per instance is $\mathcal{O}(S\,\widehat{N} + \widehat{N}^3)$.

## C.2 Molecular generation in field space with EDM

**Fields and importance sampling.** Following CORDS, a molecule with atoms at $\{\mathbf{r}_u\}_{u=1}^N$ and per-atom features $\{\phi_u\}$ (atom type logits and, when used, charge) is mapped to continuous fields on $\mathbb{R}^3$: a density $\rho(\mathbf{r})$ and feature channels $\mathbf{h}(\mathbf{r})$ built by placing normalized isotropic kernels of width $\sigma$ at each atom:

$$\rho(\mathbf{r}) = \sum_{u=1}^{N} \kappa_\sigma(\mathbf{r}-\mathbf{r}_u), \qquad \mathbf{h}(\mathbf{r}) = \sum_{u=1}^{N} \phi_u\,\kappa_\sigma(\mathbf{r}-\mathbf{r}_u).$$

At training time we *do not* operate on graphs. Instead, we discretize fields by *importance sampling* $M$ query locations from a proposal $q(\mathbf{r})$ proportional to $\rho(\mathbf{r})$, and read out the field values at those points:

$$\big\{(\mathbf{r}_i,\, \rho_{\mathrm{n}}(\mathbf{r}_i),\, \mathbf{h}_{\mathrm{n}}(\mathbf{r}_i))\big\}_{i=1}^{M},$$

where "n" denotes channel-wise normalization (coords, density, features). Densities are represented either in *log-space* ($\log \rho_{\mathrm{n}}$) or in *raw space* ($\rho_{\mathrm{n}}$), controlled by a flag; all learning is carried out directly on these fields.

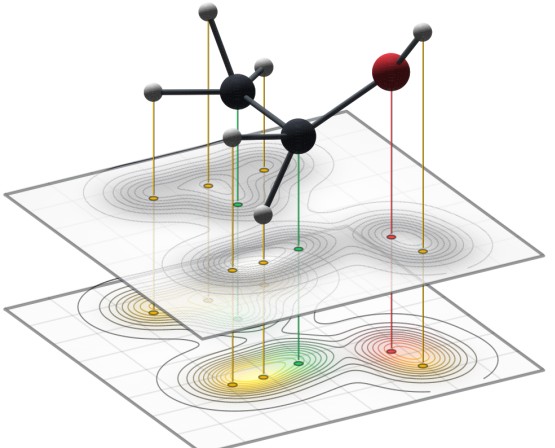

Figure 5: A molecular graph (top) is encoded with CORDS into a density field $\rho(\mathbf{r})$ (middle) and feature fields $h_k(\mathbf{r})$ (bottom), which correspond to atom types here. The number of atoms is encoded directly in the density mass, $N = \int \rho(\mathbf{r})\, d\mathbf{r}$.

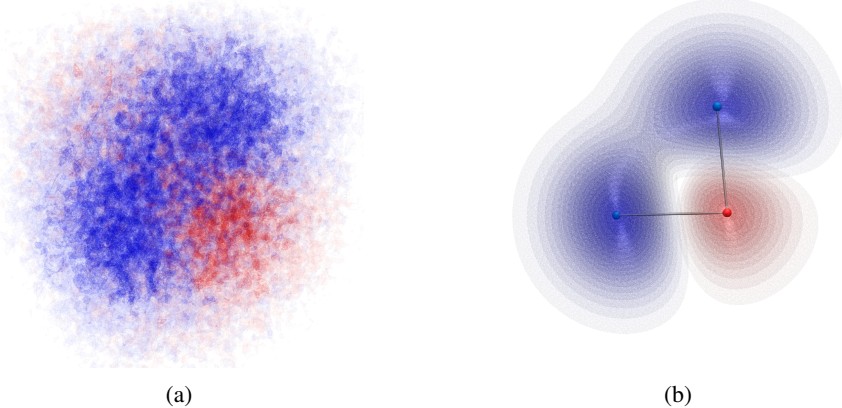

(a)             (b)

Figure 6: **Visualization of node feature fields in the continuous domain.** We depict the *node feature field* $\mathbf{h}_n(\mathbf{r})$ as 3D iso-contours over the spatial domain, with each color representing a different feature channel. **(a)** An intermediate state of $\mathbf{h}_n$ during denoising diffusion, where node features fields are still dispersed and overlap spatially. **(b)** The final denoised field after model inference, where node contributions are well-separated. Superimposed are the reconstructed discrete graph nodes obtained by applying the inverse mapping decoding, illustrating how the continuous field encodes node positions and features, enabling recovery of discrete graph structure.

**EDM preconditioning and losses.** We train a score network in the Elucidated Diffusion Models (EDM) framework to jointly denoise coordinates and field channels. Let

$$x = [\rho_n, \mathbf{h}_n] \tag{13}$$

denote the stacked field features at the sampled locations, and let $\mathbf{p}$ be the corresponding coordinates.

For each molecule, we draw a log-normal noise level by sampling $\epsilon \sim \mathcal{N}(0, 1)$ and setting

$$\sigma = \exp(\epsilon\, P_{\text{std}} + P_{\text{mean}}). \tag{14}$$

We then add isotropic Gaussian noise to all three channel families:

$$\mathbf{p}_\sigma = \text{center}(\mathbf{p}) + \sigma\, \eta_p, \qquad \rho^\star = \rho_{\text{n}} + \sigma\, \eta_\rho, \qquad \mathbf{h}^\star = \mathbf{h}_{\text{n}} + \sigma\, \boldsymbol{\eta}_h, \tag{15}$$

and define the noisy field stack

$$x^\star = [\rho^\star, \mathbf{h}^\star]. \tag{16}$$

The network operates on preconditioned inputs

$$(\mathbf{p}_{\text{in}}, x_{\text{in}}) = (c_{\text{in}}\mathbf{p}_\sigma,\ c_{\text{in}}x^\star), \tag{17}$$

and predicts residuals $(\Delta\mathbf{p}, \Delta x)$. To couple coordinate, density, and feature channels, we use a single EDM preconditioning scale based on the geometric mean

$$\sigma_{\text{data}} = (\sigma_c\, \sigma_r\, \sigma_f)^{1/3}. \tag{18}$$

The resulting EDM scalings are

$$c_{\text{skip}} = \frac{\sigma_{\text{data}}^2}{\sigma^2 + \sigma_{\text{data}}^2}, \qquad c_{\text{out}} = \frac{\sigma\, \sigma_{\text{data}}}{\sqrt{\sigma^2 + \sigma_{\text{data}}^2}}, \qquad c_{\text{in}} = \frac{1}{\sqrt{\sigma_{\text{data}}^2 + \sigma^2}}. \tag{19}$$

We form denoised estimates via

$$\widehat{\mathbf{p}} = c_{\text{skip}}\mathbf{p}_\sigma + c_{\text{out}}(\mathbf{p}_{\text{in}} - \Delta\mathbf{p}), \tag{20}$$

$$\widehat{x} = c_{\text{skip}}x^\star + c_{\text{out}}(x_{\text{in}} - \Delta x). \tag{21}$$

Training minimizes a weighted sum of MSE losses, using *per-family* $\sigma_{\text{data}}$ values (i.e., $\sigma_c, \sigma_r, \sigma_f$) to define the EDM weights. Concretely,

$$\mathcal{L} = \lambda_{\text{coords}}\, \mathcal{L}_{\text{coords}} + \lambda_\rho\, \mathcal{L}_\rho + \lambda_{\text{feats}}\, \mathcal{L}_{\text{feats}}, \tag{22}$$

where

$$\mathcal{L}_{\text{coords}} = \left\| w_c^{1/2}(\widehat{\mathbf{p}} - \text{center}(\mathbf{p})) \right\|_2^2, \tag{23}$$

$$\mathcal{L}_\rho = \left\| w_r^{1/2}(\widehat{\rho} - \rho_{\text{n}}) \right\|_2^2, \tag{24}$$

$$\mathcal{L}_{\text{feats}} = \left\| w_f^{1/2}(\widehat{\mathbf{h}} - \mathbf{h}_{\text{n}}) \right\|_2^2, \tag{25}$$

and the EDM weights are

$$w_c = \frac{\sigma^2 + \sigma_c^2}{(\sigma\, \sigma_c)^2}, \qquad w_r = \frac{\sigma^2 + \sigma_r^2}{(\sigma\, \sigma_r)^2}, \qquad w_f = \frac{\sigma^2 + \sigma_f^2}{(\sigma\, \sigma_f)^2}. \tag{26}$$

Optionally, we add a *mass regularizer* that penalizes the squared error between the predicted and true total density mass. Writing $\langle N \rangle$ for the mass computed by averaging $\rho$ over the point set, this term penalizes

$$\langle \widehat{N} \rangle - \langle N \rangle, \tag{27}$$

and it is compatible with both log-density and raw-density parameterizations.

**Sampler (Euler–Maruyama with Karras schedule).** At test time we draw initial Gaussian noise $(\mathbf{p}_0, x_0)$ and integrate the EDM Euler–Maruyama sampler along a Karras $\sigma$-ladder $t_0 = \sigma_{\text{max}} > \cdots > t_K = 0$ with optional "churn":

$$(\mathbf{p}_{k+1}, x_{k+1}) \leftarrow (\mathbf{p}_k, x_k) + (t_{k+1} - \hat{t}_k)\frac{(\mathbf{p}_k - \widehat{\mathbf{p}}_{\hat{t}_k},\ x_k - \widehat{x}_{\hat{t}_k})}{\hat{t}_k} \quad \text{with} \quad \hat{t}_k = t_k + \gamma t_k,$$

and a Heun correction on every step. If we enable message passing ($> 0$ steps), we rebuild a radius graph on the *current* coordinates after each update; translation is removed by centering coordinates at the end. The sampler produces a set of points and denoised fields $\{(\mathbf{r}_i, \widehat{\rho}_{\text{n}}(\mathbf{r}_i), \widehat{\mathbf{h}}_{\text{n}}(\mathbf{r}_i))\}_{i=1}^M$.

**Decoding back to molecules (evaluation only).** All training and sampling happen in field space. For metrics that require graphs, we apply the same decoder as in CORDS: (i) estimate $\hat{N}$ from the density mass, (ii) fit atom centers by kernel matching to $\widehat{\rho}_{\mathrm{n}}$, and (iii) reconstruct per-atom features by a weighted linear solve from $\widehat{\mathbf{h}}_{\mathrm{n}}$. When charges are modeled, they are carried as continuous channels in $\mathbf{h}$ and decoded directly.

**Evaluation Metrics.** To assess the quality of generated molecules, we report the following standard metrics:

- **Validity**: the percentage of generated samples that correspond to chemically valid molecules, i.e., those that can be parsed and satisfy valence rules.
- **Uniqueness**: the proportion of valid molecules that are unique (non-duplicates) within the generated set.
- **Atom Stability**: the percentage of atoms in each molecule whose valence configuration is chemically stable.
- **Molecule Stability**: the percentage of molecules in which *all* atoms are stable, i.e., no atom violates valence constraints.

These metrics are computed using a standardized chemistry toolkit and follow established benchmarks for QM9 generation. Together, they reflect both structural correctness and chemical diversity of the generated samples.

**Backbone and normalization.** The network acting on unordered field samples is an Erwin-based encoder–decoder (Fieldformer head), receiving *both* coordinates and field channels. We apply consistent channel normalization: coordinates are scaled by a fixed factor, features by another, and the density channel is either $\log \rho$ or $\rho$ with its own scale. All $\sigma_{data}$ values are estimated from the RMS of these *normalized* channels on training data; the EDM ladder $(\sigma_{\min}, \sigma_{\max})$ is set to yield a wide SNR range on coordinates.

RUNTIME AND SCALABILITY FOR MOLECULAR GENERATION

**Asymptotic costs.** For a batch of molecules with at most $N$ atoms, spatial dimension $d$, feature dimension $d_x$, and $M$ field samples per molecule, the CORDS encoding used during training scales as

$$\mathcal{O}\big(MNd_x\big)$$

to evaluate density and feature fields at the sampled locations. The Erwin–EDM backbone then scales linearly in $M$ (number of points), so our overall per-step complexity is $\mathcal{O}(MNd_x)$. Decoding at inference time follows the analysis in Appendix C.1: given a decoded molecule with $\hat{N}$ atoms and $S$ evaluation points, the position/feature reconstruction scales as

$$\mathcal{O}\big(S\hat{N} + \hat{N}^3\big),$$

where the $\hat{N}^3$ term comes from the small Gram solve. In our regimes ($\hat{N} \leq 29$ on QM9, moderate $\hat{N}$ on GeomDrugs), the $\hat{N}^3$ term is negligible compared to the $S\hat{N}$ kernel evaluations; the feature solve becomes dominant only when $\hat{N}$ is very large and $S$ is small, clarifying the remark in Appendix C.1.

**Measured training and inference times.** Table 3 summarizes the measured runtimes for QM9 generation on a single NVIDIA H100 GPU (batch size 128, 750k training steps). "Encode" refers to CORDS field construction during training, and "Decode" comprises position reconstruction (GMM + LBFGS) and feature recovery.

**Discussion.** Encoding is used only during training and contributes less than $1\%$ of the per-step wall-clock time relative to the diffusion backbone. Decoding is used only during sampling and adds a moderate overhead: on QM9 the CORDS decoding time is on the same order as a single pass of the generative network, with position refinement currently dominating this cost. Our prototype implementation does not yet fully vectorize the position fitting; we expect a further order-of-magnitude speedup from optimizing this step in the public release. Overall, for small molecules (QM9) and

Table 3: Approximate runtimes for QM9 molecular generation on a single H100 GPU. Times are reported per molecule.

| Phase / component | Time [ms] | Description |
|---|---|---|
| Training: NN forward+backward | 4.32 | EDM + Erwin backbone |
| Training: CORDS encode | 0.014 | atoms $\rightarrow$ fields |
| Inference: NN sampling | 58.8 | 30 EDM steps in field space |
| Decode: position reconstruction | 19.98 | GMM search + LBFGS refinement |
| Decode: feature recovery | 0.57 | atom-type / feature Gram solve |
| Decode total | 20.8 | full CORDS decoding |
| End-to-end total | 79.6 | sampling + decoding |

medium-sized drug-like molecules (GeomDrugs), the additional overhead of CORDS remains practical, while applications to very large macromolecules would require additional engineering and are left for future work.

EDM PIPELINE

```
{Pseudocode (EDM training, sampling, decoding)}
# --- Encode : atoms -> fields, sample M points ---
def rasterize_and_sample(atoms):
    rho, h = make_fields(atoms, kernel=gaussian(sigma))
    r_i ~ q(r)  rho(r)                 # importance sampling
    dens = log(rho(r_i)) if use_log_rho else rho(r_i)
    coords = r_i / norm_coords
    feats  = h(r_i) / norm_feats
    dens   = dens / norm_rho
    return coords, stack([dens, feats], -1)    # [B,M,3], [B,M,1+C]

# --- EDM training ---
for batch in loader:
    coords, feats = rasterize_and_sample(batch)
    sigma = exp(P_mean + P_std * randn([B,1]))   # per-molecule noise
    pos_noisy  = center(coords) + sigma * randn_like(coords)
    dens_noisy = feats[..., :1] + sigma * randn_like(feats[..., :1])
    h_noisy    = feats[..., 1:] + sigma * randn_like(feats[..., 1:])
    x_noisy = concat([dens_noisy, h_noisy], -1)

    # preconditioning
    c_skip, c_in, c_out = edm_scalings(sigma, sigma_data=(_c,_r,_f))
    pos_in, x_in = c_in * pos_noisy, c_in * x_noisy
    dpos, dx = FieldModel(pos_in, x_in, graph=radius_graph(pos_in)
    if mp>0 else None)
    pos_hat = c_skip * pos_noisy + c_out * (pos_in - dpos)
    x_hat   = c_skip * x_noisy   + c_out * (x_in   - dx)

    # weighted losses
    L = _coords * mse_w(pos_hat, center(coords), w_c(,_c)) \
      + _rho     * mse_w(x_hat[:,:1], feats[:,:1], w_r(,_r)) \
      + _feats   * mse_w(x_hat[:,1:], feats[:,1:], w_f(,_f))
    if _mass>0: L += _mass * (mass(x_hat[:,:1]) -
    mass(feats[:,:1]))**2
    L.backward(); opt.step(); opt.zero_grad()

# --- Sampling (EulerMaruyama + Karras ladder) ---
x, pos = randn([B,M,1+C]), randn([B,M,3]); pos = center(pos)
for t_cur, t_next in karras_schedule(_max, _min, K):
    x_hat, pos_hat = churn(x, pos, t_cur, S_churn, S_noise)
    x_d, p_d = FieldModel(c_in(t_hat)*pos_hat, c_in(t_hat)*x_hat)
    dx = (x_hat - x_d) / t_hat; dp = (pos_hat - p_d) / t_hat
    x, pos = heun_update(x, pos, dx, dp, t_cur, t_next)
    if mp>0: graph = radius_graph(pos)
pos = center(pos)

# --- Decode  (for metrics only) ---
N_hat = integral_of_density(x[:,:1])        # count from mass
t0 = fit_kernel_centers(r=pos, rho=x[:,:1], K=N_hat)
features = gram_projection(h=x[:,1:], centers=t0)
```

In the previous code, the `FieldModel` is based on Erwin, and can be summarized as follows.

```
{--- FieldModel / Erwin block ---}
# Inputs:  pos [B,M,3], feats [B,M,1+C], sigma [B,1], batch [B*M],
    cond [B,K] or [B*M,K]
# Outputs: dpos [B,M,3], dfeat [B,M,1+C]

def fieldformer_step(pos, feats, sigma, *, batch, cond=None):
    # 1)  embedding (log scaled)
    log_sigma = log(sigma) / 4.0
    log_sigma_nodes = broadcast_to_nodes(log_sigma, batch)    #
    [B*M,1]

    # 2) Encode sampled points
    z_pos   = RFF(pos.view(-1, 3))                            # coords
    z_feat  = FeatMLP(feats.view(-1, 1+C))                    # fields
    z_sigma = SigmaEmbed(log_sigma_nodes)                     #
    if cond is not None:
        z_cond = ConditionEmbed(broadcast_cond(cond, batch))
        z_in   = concat([z_pos, z_feat, z_sigma, z_cond], -1)
    else:
        z_in   = concat([z_pos, z_feat, z_sigma], -1)

    # 3) Fuse encodings
    h0 = Linear(z_in, out_dim=H)

    # 4) FiLM modulation by
    h0 = SigmaFiLM(H)(h0, log_sigma_nodes)

    # 5) Erwin/Transformer trunk over field points
    h  = main_model(h0, node_positions=pos.view(-1,3),
    batch_idx=batch)

    # 6) Prediction head  per-point residuals
    y  = PredHead(h)
    dpos, dfeat = split(y, sizes=(3, 1+C), dim=-1)

    # 7) Reshape back
    return dpos.view(B,M,3), dfeat.view(B,M,1+C)
```

### C.3   OBJECT DETECTION (MULTIMNIST)

**Goal and idea.**    We compare three detectors that differ only in *representation principle* while keeping capacity and engineering comparable: (i) a **field-based (CORDS)** detector that predicts aligned continuous fields (§3), (ii) a **YOLO-like** anchor-free detector (stride 8), and (iii) a **DETR-like** query-based detector. The YOLO/DETR baselines are deliberately *minimal* (no large-scale tricks or post-hoc stabilizers) so that the comparison focuses on the core ideas: density mass for counting (CORDS), cell-wise anchors (YOLO), and slot/query capacity (DETR).

**Dataset and splits.**    We use an on-the-fly MULTIMNIST generator (black background, no extra augmentations). Each image is $128 \times 128$, with a uniformly sampled number of digits per image. Digits are randomly rotated ($\pm 25°$) and rescaled to a side length in $[18, 42]$ pixels and pasted on the canvas with a small border margin. The training range is $N \in [1, N_{\max}]$ with $N_{\max}=15$. We report (i) in-distribution (ID) metrics on held-out images with at most $N_{\max}$ objects, and (ii) an OOD split with exactly $N_{\max}+1$ objects to probe robustness to *variable cardinality*.

**Targets and what the models predict.**    A scene is a set of discrete objects (bounding boxes of digits). Each instance is $(x, y, w, h, c)$ with center $(x, y) \in \mathbb{R}^2$, size $(w, h)$, and class $c \in \{0, \dots, 9\}$. CORDS encodes this set into fields on the image plane: $\rho(\mathbf{r})$ (density), $\rho(\mathbf{r}) \pi_k(\mathbf{r})$ (per-class mass channels), and $\rho(\mathbf{r}) \mu(\mathbf{r}) \in \mathbb{R}^2$ (size mass). The YOLO/DETR baselines predict class probabilities, boxes, and objectness/no-object in their usual forms.

**Training objective (shared outline).** All models are trained from RGB images (standard mean/std normalization) to their respective targets. For CORDS we minimize a pixel-/sample-wise MSE on the field channels plus a mass-based count penalty (as in the main paper):

$$\mathcal{L} \;=\; \mathcal{L}_{\mathrm{MSE}}(\rho, \rho\pi, \rho\mu) \;+\; \lambda\left(\hat{N} - N\right)^2, \qquad \hat{N} \;=\; \int \rho(x, y)\, dx\, dy.$$

(The model learns to make $\hat{N}$ an accurate, *differentiable* count.) For YOLO we use objectness BCE, class CE (positives), and box L1+GIoU. For DETR we use the standard Hungarian matching with class CE (with a no-object weight), L1 on boxes, and GIoU.

**Backbones and heads (capacity parity).** We keep capacities comparable ($\approx 8$M parameters total) by adjusting base widths:

- **CORDS fields**: a light ConvNeXt-like FPN (stride 8 feature map; full-resolution output), head predicts $1+K+2$ channels: $\rho$, $\rho\pi_{1:K}$, $\rho\mu$ (with $\mu \in [0,1]^2$). We also implement a tiny UNet head; both behave similarly.
- **YOLO-like**: a stride-8 CNN/ConvNeXt-lite backbone feeding a minimal anchor-free head that regresses cell-relative $(c_x, c_y, w, h)$ and predicts objectness and $K$-way class logits.
- **DETR-like**: a small CNN/ConvNeXt-lite backbone ($H/8$ or $H/16$ stride) followed by a 3-layer Transformer encoder and 3-layer decoder with $T = N_{\mathrm{max}}$ learned queries; heads predict $K+1$ class logits (incl. no-object) and a normalized box via a 3-layer MLP.

**How counting differs.** DETR is *slot-limited* (at most $T$ outputs). YOLO is *grid-limited* but flexible in count after NMS. CORDS learns $\hat{N}$ from the *mass* of $\rho$, so increasing scene density produces larger $\int \rho$ without architectural changes; decoding scales seamlessly to OOD counts.

CORDS (FIELDS) MODEL: FORWARD, TRAINING, AND DECODING

**Forward prediction.** The fields head outputs unnormalized logits which are mapped as $\rho = \mathrm{softplus}(\cdot)$, $\pi = \mathrm{softmax}(\cdot)$, $\mu = \sigma(\cdot)$, and then $[\rho, \; \rho\pi, \; \rho\mu]$.

**Training with Monte Carlo supervision.** We train against *sparsely sampled* field targets built on-the-fly to avoid full-image integrals. For each image we draw $S=4096$ points $\{\mathbf{r}_s\}$ as a mixture of uniform and importance sampling (fraction $p_{\mathrm{imp}}=0.6$); we evaluate the analytic $\rho, \rho\pi, \rho\mu$ at those points and regress with an MSE weighted by MC weights $w_s$ (optionally unbiased). This is a practical implementation of the integrals used for feature inversion in §3: uniform sampling corresponds to constant $w_s$, while importance sampling uses $w_s \propto 1/q(\mathbf{r}_s)$ with $q$ proportional to a kernel mixture around object centers (see code).

**Decoding at test time.** We decode with a simple seed-and-refine routine operating on the predicted fields (no heavy post-processing). In brief: (1) compute per-class mass maps $\rho\pi_k$; (2) infer how many seeds to extract either per-class or globally from $\sum \rho$; (3) take local maxima (optionally subpixel refinement); (4) read $\mu$ (size) and $\pi$ (class) at seed locations; (5) score seeds by conf $= \pi_k \cdot (1 - \exp(-\rho_{\mathrm{peak}}))$ and apply NMS. A fixed `per_image_topk` cap ($= N_{\mathrm{max}}$) is used for fairness.

```
{Illustrative pseudo-code (CORDS).}
def fields_forward(x):
    feat = backbone_convnext_fpn(x)           # [B,C,H/8,W/8] ->
    FPN -> [B,C,H,W]
    logits = head_1x1(feat)                    # [B, 1+K+2, H, W]
    rho = softplus(logits[:, :1])              # density >= 0
    pi  = softmax(logits[:, 1:1+K], dim=1)     # per-class probs
    mu  = sigmoid(logits[:, 1+K:1+K+2])        # size in [0,1]^2
    return torch.cat([rho, rho*pi, rho*mu], dim=1)

def fields_decode(maps, K, H, W, alpha=1.0, nms_iou=0.5, topk=15):
    rho, cls_mass, size_mass = split(maps)     # [1], [K], [2]
    pi  = normalize(cls_mass, by=rho)          # pi = cls_mass / rho
    mu  = clamp(size_mass / rho, 0, 1)
    # how many per class? use density mass:
    Nk = round(alpha * (rho[None,:,:] * pi).sum((-2,-1)))   # [K]
    dets = []
    for k in range(K):
        seeds = topk_local_maxima((rho*pi[k]).squeeze(0), Nk[k])
        # optional subpixel refine (soft-argmax in a small window)
        wh    = bilinear_sample(mu, seeds)
        conf  = bilinear_sample(pi[k], seeds) * (1 -
    exp(-bilinear_sample(rho[0], seeds)))
        boxes = seeds_to_xyxy(seeds, wh, H, W)
        dets += nms_select(boxes, conf, class_id=k, iou=nms_iou)
    return prune_topk(dets, topk)
```

YOLO-LIKE BASELINE (ANCHOR-FREE, STRIDE 8)

**Backbone/head and parameterization.** A tiny CNN or ConvNeXt-lite backbone produces a stride-8 feature map. The head predicts for each cell: objectness, class logits over $K$ digits, and a cell-relative box $(c_x, c_y, w, h)$ with

$$c_x = \tfrac{g_x + \sigma(t_x)}{W_s},\ c_y = \tfrac{g_y + \sigma(t_y)}{H_s},\quad w = \sigma(t_w)^2,\ h = \sigma(t_h)^2,$$

where $(g_x, g_y)$ is the integer cell coordinate and $H_s, W_s$ are stride-8 sizes. This prevents "teleporting" boxes from distant cells.

**Assignment and loss.** We assign each GT to its nearest cell (or the $k$ nearest; $k = 1$ by default). Losses: $\mathcal{L}_{\text{obj}} = \text{BCEWithLogits}$ on all cells with negative down-weight, $\mathcal{L}_{\text{cls}} = \text{CE}$ on positives (optional label smoothing), $\mathcal{L}_{\text{box}} = \text{L1}$ on $(c_x, c_y, w, h)$ + GIoU in pixels.

**Eval.** Scores are $\text{obj} \cdot \max_k p_k$. We apply NMS (class-agnostic or per-class) and cap to `per_image_topk`.

```
{Illustrative pseudo-code (YOLO).}
def yolo_forward(x):
    f = backbone_stride8(x)                     # [B,C,Hs,Ws]
    logits_cls, logits_obj, t_box = head(f)     # [B,K,Hs,Ws],
    [B,1,Hs,Ws], [B,4,Hs,Ws]
    cx, cy, w, h = decode_cell_relative(t_box)  # normalized to [0,1]
    return dict(cls_logits=flatten(logits_cls),
    obj_logits=flatten(logits_obj),
                pred_boxes=flatten(stack([cx,cy,w,h]))))

def yolo_decode(out, H, W, conf_thr=0.25, nms_iou=0.4, topk=15):
    prob = softmax(out["cls_logits"], dim=-1)   # [B,N,K]
    obj  = sigmoid(out["obj_logits"])           # [B,N]
    scores, labels = prob.max(-1)               # [B,N], [B,N]
    score = obj * scores
    boxes_xyxy = cxcywh_to_xyxy_norm(out["pred_boxes"]) *
    [W-1,H-1,W-1,H-1]
    keep = score >= conf_thr
    dets = nms_per_class_or_agnostic(boxes_xyxy, score, labels,
    iou=nms_iou)
    return prune_topk(dets, topk)
```

DETR-LIKE BASELINE (MINIMAL)

**Backbone/transformer.** A compact backbone produces a $d$-dimensional feature map, augmented with coordinate channels and sine positional encodings. A 3-layer encoder and 3-layer decoder operate on $T$ learned queries. We set $T = N_{\max}$ to reflect a "budget" comparable to the other models.

**Matching, loss, and eval.** Hungarian assignment (SciPy) is used to match predictions to GT; if SciPy is unavailable a greedy fallback is used. Losses: class CE with a reduced weight for the no-object class, L1 on boxes (in $(c_x, c_y, w, h)$), and GIoU in pixels. At inference we compute scores as $\text{score} = (1 - p_{\text{noobj}}) \cdot \max_k p(c{=}k)$, optionally apply NMS, and cap to `per_image_topk`.

```
{Illustrative pseudo-code (DETR).}
def detr_forward(x):
    f = backbone(x)                             # [B,C,Hs,Ws]
    pos = sine_posenc(f); coord = coord_channels(f)
    src = project(cat([f, coord])) + pos        # [B,C,Hs,Ws]
    S,B,C = (Hs*Ws), x.size(0), src.size(1)
    mem = encoder(flatten(src) + flatten(pos))  # [S,B,C]
    tgt = zeros(T,B,C); qpos = query_embed(T,B,C)
    hs  = decoder(tgt + qpos, mem + flatten(pos))# [T,B,C]
    return dict(pred_logits=class_head(hs),
    pred_boxes=sigmoid(box_mlp(hs)))

def detr_decode(out, H, W, conf_thr=0.4, nms_iou=0.6, topk=15):
    prob = softmax(out["pred_logits"], dim=-1)  # [..., K+1]
    p_no = prob[..., K]; p_cls, labels = prob[...,:K].max(-1)
    score = (1.0 - p_no) * p_cls
    boxes_xyxy = cxcywh_to_xyxy_norm(out["pred_boxes"]) *
    [W-1,H-1,W-1,H-1]
    keep = score >= conf_thr
    dets = optional_nms(boxes_xyxy, score, labels, iou=nms_iou)
    return prune_topk(dets, topk)
```

HYPERPARAMETERS AND FAIRNESS GUARD

**Common.** Image size $128 \times 128$; classes $K{=}10$; $N_{\max}{=}15$. Optimizer AdamW (lr $1 \times 10^{-4}$, weight decay $5 \times 10^{-4}$), batch size 128, 200 epochs. We cap detections to `per_image_topk`$= 15$ in *all* methods and allow optional class-agnostic NMS for fairness.

**CORDS (fields).** Backbone: Light ConvNeXt-FPN (stride 8). Head width "base" $= 64$ (chosen so the total params $\approx$ YOLO/DETR). Density activation: `softplus` (or `softplus0`) to ensure $\rho \geq 0$. Sampling: $S{=}4096$ points/image with importance fraction $p_{\text{imp}}{=}0.6$; Gaussian kernel bandwidth $\sigma_{\text{norm}}{=}0.02$ (fraction of $\min\{H, W\}$). Training loss weights: $w_\rho{=}4$, $w_{\text{cls}}{=}1$, $w_{\text{size}}{=}2$. Optional weak count supervision on a random fraction of the batch (weight $w_{\text{count}}{=}0.5$; off by default). At decode: `seed_radius`$= 0.03$, `decode_alpha`$= 1.0$, NMS IoU $= 1.0$ (disabled unless stated), and `per_image_topk`$= 15$. We rescale training fields by a constant $R$ for numerical stability (`rho_rescale`$=10.0$) and undo it before decoding and metrics.

**YOLO-like.** Backbone: tiny CNN or ConvNeXt-lite to stride 8 feature map with $d_{\text{model}}{=}256$. Loss weights: $w_{\text{obj}}{=}1$, $w_{\text{cls}}{=}1$, $w_{\text{box-L1}}{=}2$, $w_{\text{GIoU}}{=}2$. No-object down-weight $= 0.5$. Label smoothing $= 0.0$ (unless specified). Assignment: center cell (or $k$-nearest cells with $k{=}1$ by default). Inference: confidence threshold 0.25; NMS IoU 0.40; optional class-agnostic NMS; `per_image_topk`$= 15$.

**DETR-like.** Backbone: tiny CNN or ConvNeXt-FPN; Transformer $d{=}256$, $n_{\text{heads}}{=}8$, #enc/dec layers $= 3/3$, FFN 1024. Queries $T = N_{\max}$ unless noted. Loss weights: class 1.0 (no-object coefficient 0.5), $L_1$ on boxes 5.0, GIoU 2.0. Eval: score $= (1 - p_{\text{noobj}}) \cdot \max_k p_k$; threshold 0.40; optional NMS with IoU 0.60; `per_image_topk`$= 15$.

**Metrics.** We report AP at IoU $0.50, 0.75, 0.90$, the mean over $[0.50{:}0.95]$ in steps of $0.05$ (mAP$_{50:95}$), and a headline mAP$_{50:75}$. *Count MAE* is $|\#\text{preds} - \#\text{GT}|$ averaged over images, where $\#\text{preds}$ is (i) $\int \rho$ for CORDS and (ii) the number of post-NMS detections for YOLO/DETR (both capped to `per_image_topk` for fairness).

**OOD protocol.** For the OOD split we set the test cardinality to $N{=}N_{\max}{+}1$. Query-based models necessarily under-count when $T{<}N$. In contrast, the field-based model increases $\int \rho$ naturally with scene density and decodes with the same routine, without changing the network or its capacity.

### C.4 Simulation-based inference for FRBs (implementation)

**Problem setting.** We consider 1D photon-count light curves with Poisson noise generated by a variable number of transient components (FRB bursts). Each component is parameterized by onset time $t_0$, amplitude $A$, rise time $\tau_{\text{rise}}$, and skewness skew. The latent cardinality $N$ is unknown and changes per observation. Our goal is amortized posterior inference $p\big(\{(t_{0,u}, A_u, \tau_u, \text{skew}_u)\}_{u=1}^N \mid \ell\big)$.

**Representation.** As in CORDS, we map sets of components to continuous *fields* on the time axis $t \in [0, 1]$: a density $\rho(t)$ that places unit mass around each $t_0$, and a feature field $\mathbf{h}(t)$ that carries the remaining parameters over the same support. We show this in Figure 7 Practically, $\rho(t)$ and $h$-channels are built by convolving Dirac impulses with normalized Gaussians of bandwidths $\sigma_\rho$ and $\sigma_{\text{feat}}$, respectively, so that $\int \rho \, dt = N$ when evaluated continuously. We discretize on a uniform grid of $T$ points (here $T{=}1000$) and optionally downsample by average pooling to $T_{\text{eff}}{=}T/\text{downsample}$.

FRB Timeseries and Fields

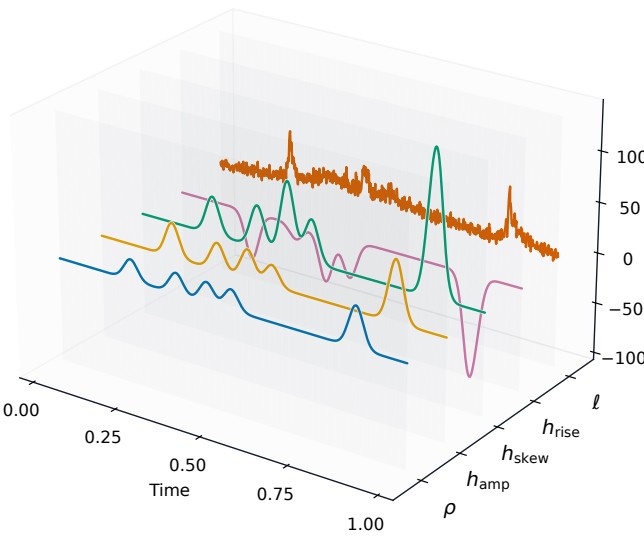

Figure 7: An example of a lightcurve $\ell$, and corresponding burst components encoded into density and feature fields in the time domain.

**Conditioning and model.** Given a light curve $\ell \in \mathbb{R}^T$, we standardize it (per-sample or using a precomputed global normalizer) and append it as a conditioning channel. The network is a 1D conditional residual stack (`CondTemporalNet`) with grouped norms, safe dilations, optional lightweight self-attention, and a learned sinusoidal time embedding used by all residual blocks. The model maps *fields* $+ \ell$ to *fields* (same channel count).

**Flow matching objective (FMPE).** We use flow matching for SBI Dax et al. (2023). For each training pair we sample $t \in (0,1)$ (power-law schedule), form a noisy interpolation $f_t = \mu_t + \sigma_t \varepsilon$ with $\mu_t = t\,x_1$ and $\sigma_t = 1 - (1 - \sigma_{\min})t$ (here $x_1$ are target fields and $\varepsilon \sim \mathcal{N}(0, I)$), and supervise the time-dependent velocity field $u_t$ that transports $f_t$ to $x_1$:

$$u_t = \frac{x_1 - (1 - \sigma_{\min})\, f_t}{1 - (1 - \sigma_{\min})\, t}, \qquad \mathcal{L}_{\text{FMPE}} = \big\| v_\theta(f_t, \ell, t) - u_t \big\|_2^2.$$

This yields an amortized posterior over fields $p(\rho, \mathbf{h} \mid \ell)$. We also maintain an optional EMA of parameters for stable sampling.

**Decoding back to components ($\Psi$).** Given predicted fields on the grid:

1. **Cardinality $\hat{N}$.** Estimate $\hat{N} = \text{round}\big(\Delta t \sum_i \rho(t_i)\big)$, where $\Delta t$ is the grid spacing. Because $\rho$ comes from a smooth network and finite grids, the sum is rarely an integer; we round to the nearest integer ($\pm 0.5$ rule). This gives the number of kernels to fit next.

2. **Onset times $t_0$.** Extract $\hat{N}$ seeds by greedy NMS on $\rho$ with a minimum separation of roughly $2\sigma_\rho$ (in pixels/time-steps), refine each seed by a quadratic sub-pixel step, and finally run a short LBFGS that minimizes $\|\sum_u \kappa(\cdot - t_{0,u}) - \rho\|_2^2$ w.r.t. $\{t_{0,u}\}$, with $\kappa$ the normalized Gaussian used in $\Phi$.

3. **Features $(A, \tau_{\text{rise}}, \text{skew})$.** Solve the *weighted normal equations* (Gram projection) $GX = B$ with $G_{uv} = \sum_i w_i\, \kappa(t_i - t_{0,u})\, \kappa(t_i - t_{0,v})$, $B_{u:} = \sum_i w_i\, \kappa(t_i - t_{0,u})\, \mathbf{H}(t_i)^\top$, where $\mathbf{H}$ stacks the feature channels of $\mathbf{h}$ and $w_i$ are quadrature/MC weights. On a uniform grid, $w_i = \Delta t$ (constant); with importance sampling, $w_i = \frac{1}{S\,q(t_i)}$ are standard unbiased MC weights. We add a small Tikhonov term to $G$ and solve for $X$.

**Posterior use at test time.** To estimate $p(N \mid \ell)$ we sample many fields from the learned flow, decode each to $\hat{N}$, and histogram the outcomes. For predictive curves (Fig. 4) we decode each sample to parameters, render a Poisson rate curve via the FRB generator, and aggregate (median and quantiles). For corner plots we restrict to samples where $\hat{N}$ equals the ground-truth count, align components by nearest-onset in 1D, and visualize the empirical posterior over aligned parameters.

PSEUDOCODE (TRAINING, SAMPLING, DECODING)

```
{Pseudocode}
# --- Encode : set -> 1D fields ---
def encode_fields(params, time, sigma_rho, sigma_feat):
    # rho(t)= N(t;t0,); h_k(t)= _k N(t;t0,f)
    rho, H = 0, []
    for u in range(N_max):
        m = params.active_mask[u]                   # 0/1
        k_r = gauss(time - params.t0[u], sigma_rho)
        k_f = gauss(time - params.t0[u], sigma_feat)
        rho += m * k_r
        feats_u = stack([params.ampF[u], params.riseF[u],
    params.skew[u]])
        H.append(m * feats_u * k_f)
    H = sum(H)                                       # [3, T]
    return stack([rho, H[0], H[1], H[2]], axis=0)    # [C, T]

# --- FMPE training step ---
def fmpe_step(model, fields_tgt, lightcurve, sigma_min, alpha_t):
    B, C, T = fields_tgt.shape
    t = sample_power(alpha=alpha_t, shape=[B])        # (0,1)
    t_ = t[:,None,None]
    mu_t    = t_ * fields_tgt
    sigma_t = 1.0 - (1.0 - sigma_min) * t_
    eps     = randn_like(fields_tgt)
    f_t     = mu_t + sigma_t * eps
    u_t     = (fields_tgt - (1.0 - sigma_min) * f_t) / (1.0 - (1.0
    - sigma_min) * t_)
    v_hat   = model(f_t, cond=standardize(lightcurve), time=t *
    (T_time_embed-1))
    return mse(v_hat, u_t)

# --- Sampling + decode  ---
def sample_and_decode(model, lightcurve, S, ode_steps, cfg):
    cond = standardize(lightcurve)[None,None,:]        #
    [1,1,T_eff]
    F = integrate_ode(model, f0=randn([S,C,T_eff]),
    cond=cond.repeat(S,1,1),
                      steps=ode_steps)
    Ns, t0s, Xs = [], [], []
    for s in range(S):
        rho = clamp_min(F[s,0], 0)
        N_hat = round(delta_t * rho.sum())             # mass ->
    count
        seeds = nms_on_1d(rho, K=N_hat,
    min_sep=2*sigma_rho/delta_t)
        t0_ref = lbfgs_refine(seeds, rho, sigma_rho)
        X = solve_weighted_gram(time, H=F[s,1:4], t0=t0_ref,
                                sigma_feat=sigma_feat, w=delta_t)
        Ns.append(N_hat); t0s.append(t0_ref); Xs.append(X)
    return Ns, t0s, Xs
```

ARCHITECTURAL DETAILS

**Temporal backbone (`CondTemporalNet`).** A 1D conv stack with grouped norms and residual connections; dilations grow geometrically but are capped by the effective signal length to keep reflect padding valid. Optional squeeze–excite improves channel calibration. Self-attention blocks can be inserted at chosen depths. Inputs are $\big[\texttt{fields}, \ell\big]$ (concatenated along channels), outputs are residual updates in field space (same channel count).

**Channels.** We use $C=4$ channels by default: $\rho$, $h_{\mathrm{amp}}$, $h_{\mathrm{rise}}$, $h_{\mathrm{skew}}$. The amplitude and rise channels are logarithmic by default (base-10); an optional $h_{t_0}$ channel can be added.

**Normalization.** Either per-sample standardization (zero mean, unit variance per sequence) or a global normalizer estimated over a large pool of simulated field tensors (per-channel mean/-variance). The light-curve condition is always standardized.

HYPERPARAMETERS (USED IN ALL REPORTED FRB RESULTS)

- **Simulator.** $T=1000$ points, background $y_{\mathrm{bkg}}=5$. $N_{\max}=6$ (during training, $N \sim \mathrm{Unif}\{1,\ldots,N_{\max}\}$). Priors: $t_0 \sim \mathrm{Unif}(0.2, 0.8)$, $\log_{10} A \sim \mathrm{Unif}(1, 2.477)$, $\log_{10} \tau_{\mathrm{rise}} \sim \mathrm{Unif}(-3, -0.222)$, skew $\sim \mathrm{Unif}(1, 6)$.

- **Fields.** $\sigma_\rho=0.01$, $\sigma_{\mathrm{feat}}=0.015$ (on $[0,1]$); optional downsample factor $\in \{1, 2, \ldots\}$.

- **Model.** Base channels 192–384 (we use 192 for the main runs), 8–12 residual blocks, dilation base 2, group norm with 8 groups, optional SE (reduction 8), optional attention heads $= 8$ at a few blocks.

- **FMPE.** $\sigma_{\min}=0.01$, $t \sim \mathrm{power}(\alpha_t=0)$ (uniform), ODE steps $= 250$ for sampling, init Gaussian scale $= 1.0$.

- **Optimization.** AdamW, lr $2 \times 10^{-4}$ with cosine decay to $2 \times 10^{-6}$, batch size 128, grad-clip at 1.0, EMA decay 0.999 (activated after 1000 steps).

- **Posterior evaluation.** $S \in [200, 1024]$ samples per observation, mini-batches of size 32–64 for the sampler.

PRACTICAL NOTES AND DIAGNOSTICS

**Counting and rounding.** On a uniform grid, $\int \rho \, dt$ is approximated by $\Delta t \sum_i \rho(t_i)$. Because the encoder/decoder operate on smoothed fields and finite $T$, the sum is rarely exactly an integer. We round to the nearest integer ($\pm 0.5$ rule). This determines *how many kernels* we fit in the subsequent location/feature steps; without it we would not know how many components to decode.

**Feature reconstruction and weights.** The Gram solve above is a discrete version of the integral equations in Appendix A. With a uniform grid the weights $w_i$ equal $\Delta t$ (constant). Under importance sampling (not used in our FRB runs but supported by the decoder), the same equations hold with unbiased weights $w_i = 1/(S \, q(t_i))$. This is the practical Monte-Carlo implementation of the feature inversion integrals.

**Peak finding and refinement.** We use greedy 1D NMS with a minimum separation proportional to $2\sigma_\rho$ (conservative for overlapping kernels), then a one-step quadratic sub-pixel adjustment on $\rho$, and finally a short LBFGS that directly minimizes the $\rho$ reconstruction error w.r.t. $\{t_0\}$.

**Evaluation protocol.** For each observation we (i) sample posterior fields and decode to get $p(N \mid \ell)$; (ii) produce predictive light-curve quantiles by rendering the generator at decoded parameters; and (iii) make corner plots only for samples with $\hat{N}$ matching the ground-truth $N$ (alignment by nearest $t_0$ on the line). We save panels, $p(N)$ histograms, median fields, and CSV/NPZ dumps for offline analysis (see code paths under `FRB_results/`).

**Ablations.** We implemented a DDPM objective for sanity but report *only* flow-matching (FMPE) results in this work; switching to DDPM leaves the rest of the pipeline unchanged.

```
{Minimal end-to-end sketch}
# --- Training (FMPE) ---
for step in range(steps_per_epoch * epochs):
    y_counts, params, _ = simulator.sample_batch(B)
    fields = encode_fields(params, time, , feat)           #
    fields_n = normalize(fields, mode=field_norm)
    y_n      = normalize(avg_pool(y_counts, downsample))
    loss = fmpe_step(model, fields_n, y_n, sigma_min=0.01,
    alpha_t=0.0)
    loss.backward(); clip_grad(1.0); opt.step(); opt.zero_grad()
    if use_ema: ema.update(model)

# --- Posterior sampling for one observation ---
with torch.no_grad():
    cond_y = normalize(avg_pool(y_obs, downsample))
    F_samples = fmpe_sample(ema_or_model, cond_y, steps=250) #
    integrate ODE
    Ns, t0s, Xs = [], [], []
    for F in F_samples:
        N_hat, t0_ref, X = decode_1d(F, time_eff, , feat) #
        Ns.append(N_hat); t0s.append(t0_ref); Xs.append(X)    #
    p(N|), parameters
```

## D  ADDITIONAL EXPERIMENTS

### D.1  PREDICTING VARIABLE NUMBER OF LOCAL MAXIMA

To showcase a more general and abstract mathematical task, we consider detecting local maxima of a scalar function $f : \mathbb{R}^2 \to \mathbb{R}$ from irregularly sampled observations. The number of peaks is unknown and varies per sample, which poses challenges for models with fixed-size outputs. A visualization of this problem can be seen in Figure 8.

We can easily cast this problem into our framework, by treating local extremas of the function as discrete objects, with positions corresponding to peak locations. Applying the CORDS encoding, the set of local maxima is transformed into a continuous node density field. The model's task is to predict this field from irregular samples of $f$, after which the decoding recovers both the number and coordinates of peaks. A prediction is labeled as *correct* if it recovers the exact number of peaks, with each predicted peak lying within an $\varepsilon$-neighbourhood of its true position. Since no straightforward baselines address this specific setup, our goal is not to outperform existing methods, but to showcase how our framework naturally handles variable cardinality and infers structured information from sparse observations.

We generate fully–annotated training examples by drawing a *single* realisation

$$f(\mathbf{r}) = \alpha\sqrt{\frac{2}{D}} \sum_{d=1}^{D} \cos(\mathbf{w}_d^\top \mathbf{r} + b_d), \quad \mathbf{r} \in [-3, 3]^2,$$

from a Gaussian process with squared–exponential kernel

$$k(\mathbf{r}, \mathbf{r}') = \alpha^2 \exp\left(-\|\mathbf{r} - \mathbf{r}'\|_2^2 / 2\ell^2\right)$$

approximated by $D = 150$ Random Fourier Features (RFF). Unless stated otherwise we use amplitude $\alpha = 1.5$ and length-scale $\ell = 0.9$.

**Cosine envelope.** To avoid pathological peaks on the domain boundary we multiply the raw field by a separable taper $E(\mathbf{r}) = e_x(x) e_y(y) \in [0, 1]$ with margin $m = 0.8$, where $e_x(x) = \frac{1}{2}\left[1 - \cos\left(\pi \operatorname{clip}(|x| - (3 - m), 0, m)/m\right)\right]$ (and analogously for $e_y$). The envelope smoothly decays to 0 in a 0.8-wide frame, guaranteeing that all local maxima lie strictly inside the open square $(-3 + m, 3 - m)^2$.

**Ground-truth peaks.** We sample a $181 \times 181$ grid, apply the envelope, and locate peaks with `peak_local_max(threshold_rel = 0.05, min_distance = 3)`. For each example we keep at most $M = 50$ peaks, padding with zeros if $K < M$.

**Training samples.** From the same GP realisation we draw $P = 4096$ i.i.d. points $\{\mathbf{r}_i\}_{i=1}^P$ uniformly from the domain. Their coordinates, the scalar value $f(\mathbf{r}_i)$, and the list of peaks

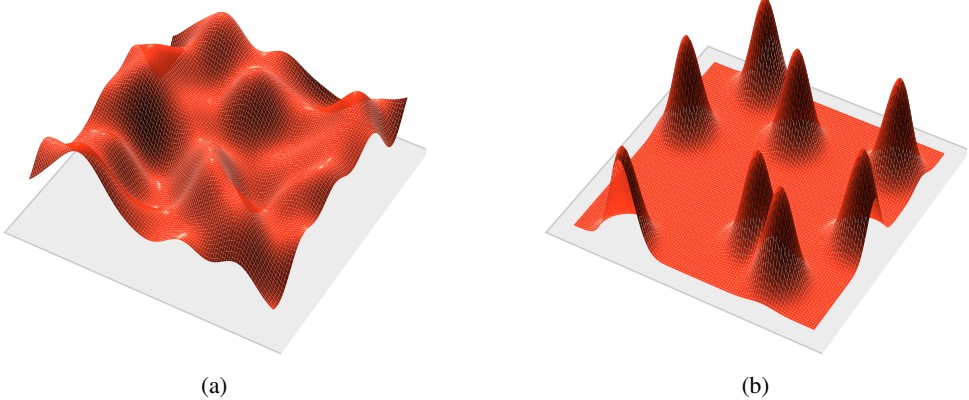

(a)            (b)

Figure 8: **(a)** Visualization of a two-dimensional input function as a surface plot. **(b)** The corresponding density field $\rho(x, y)$, obtained by interpreting the function's local extrema as discrete objects and applying the CORDS encoding.

constitute one training instance. During optimisation we randomly subsample $K = 2048$ of the $P$ points to form a mini-batch.

**Validation splits.** For robustness we evaluate on GP length-scales $\ell \in \{0.9, 1.1\}$, keeping the same envelope and amplitude.

**Accuracy criterion.** Given the set of predicted maxima $\hat{\mathcal{P}}$ and ground-truth maxima $\mathcal{P}$, we greedily assign each $\hat{p} \in \hat{\mathcal{P}}$ to its closest $p \in \mathcal{P}$. A sample is *correct* iff $|\hat{\mathcal{P}}| = |\mathcal{P}|$ *and* all assigned pairs satisfy

$$\|\hat{p} - p\|_2 \leq \varepsilon, \quad \varepsilon = \frac{\Delta_x}{T}, \quad \Delta_x = \max_t x_t - \min_t x_t, \quad T = 25.$$

with the default domain ($\Delta_x = 6$) this yields $\varepsilon = 0.24$.

**Additional training details.** All models were trained on a single NVIDIA A100 GPU using a batch size of 96. Each model took approximately 5 days to train across all experiments. We employed the AdamW optimizer with a learning rate of $5 \times 10^{-5}$, coupled with a cosine annealing schedule that reduced the learning rate to a minimum of $1 \times 10^{-6}$ over the course of training.

We observe slightly higher accuracy on the longer length scale $\ell = 1.1$, which we attribute to the smoother underlying function having fewer local maxima. This setup intentionally evaluates generalisation: the model is trained on high-frequency fields ($\ell = 0.9$) and tested on both the same and smoother fields ($\ell = 1.1$) to assess robustness across varying levels of peak density.

| $\ell$ | Accuracy (2048 points) |
|-----|-----|
| 0.9 | 92.7% |
| 1.1 | 94.2% |

Table 4: **Local Maxima Prediction Accuracy.** Accuracy of local maxima detection on different GP length scales. The model was trained on $\ell = 0.9$ using 2048 sampled points per example.

### D.2   QM9 PROPERTY REGRESSION

With CORDS, we predict molecular properties directly from the continuous field representation, without decoding back to discrete molecular graphs. Since per-node features are not needed, predictions are obtained by pooling the final representation produced by the Erwin model. We compare CORDS against representative GNN baselines (e.g., DimeNet++, SEGNN) on standard QM9 regression tasks. The aim here is not to surpass highly specialized architectures, but to show that continuous field-based representations already achieve competitive performance in this well-established domain. Results are reported in Table 5.

Table 5: QM9 regression results across different models. .

| Model | $\alpha$ | $\Delta\varepsilon$ | $\varepsilon_\mathrm{H}$ | $\varepsilon_\mathrm{L}$ | $\mu$ | $C_v$ |
|---|---|---|---|---|---|---|
| NMP | 0.092 | 69 | 43 | 38 | 0.030 | 0.040 |
| Schnet | 0.235 | 63 | 41 | 34 | 0.033 | 0.033 |
| Cormorant | 0.085 | 61 | 34 | 38 | 0.038 | 0.026 |
| L1Net | 0.088 | 68 | 46 | 45 | 0.038 | 0.030 |
| LieConv | 0.084 | 49 | 30 | 25 | 0.032 | 0.038 |
| DimeNet++* | 0.044 | 33 | 25 | 20 | 0.030 | 0.030 |
| TFN | 0.223 | 58 | 40 | 38 | 0.064 | 0.104 |
| SE(3)-Tr. | 0.142 | 53 | 35 | 33 | 0.051 | 0.054 |
| EGNN | 0.071 | 48 | 29 | 25 | 0.029 | 0.031 |
| PaiNN | 0.045 | 45 | 27 | 20 | 0.012 | 0.024 |
| SEGNN | 0.060 | 42 | 24 | 21 | 0.023 | 0.031 |
| **CORDS** | 0.085 | 50 | 32 | 30 | 0.086 | 0.039 |

**Resampling as a strong regularizer.** Molecules are encoded to fields, and we have two options: either resample fields using importance sampling at each training step, or for each molecule, evaluate sampled fields once and save them. To evaluate the role of sampling as a form of regularization, we compared two variants of CORDS on QM9 regression: one in which spatial evaluation points are resampled at each epoch (our default), and another in which the field encoding is computed once per graph and fixed throughout training. As shown in Table 6, disabling resampling leads to a dramatic increase in MAE across all targets, more than doubling the error in most cases. This confirms that stochastic sampling during training acts as a strong regularizer, promoting generalization by exposing the model to diverse field realizations. Conceptually, this is consistent with the interpretation of the model as learning an underlying continuous function that is only ever observed through a finite sampling process. Without resampling, the model risks overfitting to a specific discretization. With resampling, however, we gain robustness to spatial variation—enabling the use of large models (100M+ parameters) even on small datasets like QM9. In Table 5 we show the results on all targets compared to other baselines, with resampling at each training step.

| **QM9 Regression results (CORDS only)** | | | | | | | |
|---|---|---|---|---|---|---|---|
| **Model** | **Resample** | $\alpha$ | $\Delta\varepsilon$ | $\varepsilon_\mathrm{H}$ | $\varepsilon_\mathrm{L}$ | $\mu$ | $C_v$ |
| CORDS | ✓ | 0.085 | 50 | 32 | 30 | 0.086 | 0.039 |
| CORDS | ✗ | 0.350 | 99 | 72 | 70 | 0.240 | 0.142 |
| $\Delta$ (%) | – | +311.8% | +98.0% | +125.0% | +133.3% | +179.1% | +264.1% |

Table 6: Comparison of CORDS performance on QM9 regression with and without resampling of evaluation points. The third row shows the relative increase in error when disabling resampling.

# E   EXTENSIVE RELATED WORK

**Neural fields and continuous representations.** Neural fields, or implicit neural representations, model data as continuous functions of coordinates. Early works like DeepSDF (Park et al., 2019), NeRF (Mildenhall et al., 2020), and SIREN (Sitzmann et al., 2020) established neural fields as flexible signal representations across 3D shapes and visual data. Building on this, Functa and COIN++ (Dupont et al., 2022; 2023) explored generative modeling and cross-modal compression via neural fields. More recently, Generative Neural Fields (You et al., 2023) and Probabilistic Diffusion Fields (Zhuang et al., 2023) extended these ideas to scalable generative modeling of continuous signals. Equivariant Neural Fields (ENFs) (Wessels et al., 2025) further enhance neural fields with geometry-aware latent variables, enabling steerable, equivariant representations that support fine-grained geometric reasoning and efficient weight-sharing.

**Smoothed Fields in Computational Physics.** The formalism for continuous graph representations that we develop here bears similarities to classical methods in computational physics, such as smoothed-particle hydrodynamics (SPH) and particle-in-cell (PIC) methods (Price, 2005; Rosswog, 2009). These approaches compute scalar, vector, or tensor fields by convolving particles with smoothing kernels, typically for force computation and simulation tasks.

## LLM USAGE

Large language models (LLMs) were used to revise sentences and correct grammar, to generate visualization code for some figures, and to assist with the implementation of the MULTIMNIST dataset and corresponding baseline methods. All conceptual contributions, experiment design, analysis, and the writing of original content were carried out by the authors.

