# OpenReview forum: "CORDS - Continuous Representations of Discrete Structures"
_ICLR.cc/2026/Conference — ICLR 2026 Poster_

### Official Review · Reviewer_7wBB · 2025-10-29

**Soundness:** 1
**Presentation:** 1
**Contribution:** 1
**Rating:** 2
**Confidence:** 4

**Summary:**

The paper claims to develop CORDS (continuous representation of discrete structures), which provides a "bijective" representation that maps sets of spatial objects with features to continuous "density" and "feature fields".

**Strengths:**

None.

**Weaknesses:**

The major problem with this paper is that it uses some jargon and terms that are not mathematically defined, nor are they standard in machine learning research. This hinders the ability of a reviewer to fully understand and judge this paper. For example, starting from the first sentence of the Abstract, what does "number" mean when the paper says "their number in advance"? The Abstract talks about existing methods in the Abstract, but no problem has been laid out. The Abstract then talks about a challenge that is not substantiated.  What does "variable cadinality" mean? What does "bijective representing" mean? What is "Feature field"? These are all non-standard terminologies that I have not seen before.

**Questions:**

What is the mathematical/ML problem that the paper is solving?

**Details Of Ethics Concerns:**

I have concerns about this paper's excessive reliance on AI, as I explained in the Summary section.

---

> ### Author Response · Authors · 2025-11-19
>
> The reviewer finds several terms in the abstract unclear ("their number in advance", "variable cardinality", "bijective representation", "feature field"), feels that the mathematical / ML problem is not clearly stated, and raises an ethics concern about "excessive reliance on AI".
>
> ---
>
> **Q1: What is the mathematical / ML problem that the paper is solving?**
>
> We study ML tasks that involve reasoning about sets of variables, where the size of the set can vary from example to example. In the introduction, we give several concrete domains where this type of problem appears. The problem is stated in the first two paragraphs of Section 1 and formalized in Section 3 and Appendix A, where we define the set space, the field space, and the corresponding forward and inverse maps.
>
> ---
>
> **Q2: Clarifying the terminology in the abstract**
>
> We feel that we have inadvertently confused the reviewer with wording that we ourselves felt was innocuous. In particular:
>
> - "Their number in advance" simply means "number of objects" and appears in the first paragraph of Section 1.
> - "Variable cardinality" (unknown set size $N$) is described in Section 1 when introducing the data model.
> - "Bijective representation" is the invertible pair of maps defined in Section 3 and analyzed in Appendix A.
> - "Feature field" is defined explicitly in Eq. (1) and discussed throughout Section 3.
>
> We will revise the abstract to avoid language that could be confusing.
>
> ---
>
> **Q3: Ethics concern and AI tools**
>
> The reviewer does not specify a concrete integrity issue, but we clarify that we included an explicit "LLM usage" note at the end of the paper. There we state that LLMs were used only to revise parts of the text, and to aid in recreating baselines and dataset code for the MultiMNIST experiment. We cannot think of anything that could reasonably be considered excessive about this usage.

---

> > ### Comment · Reviewer_7wBB · 2025-11-22
> > **Thanks for the clarification.**
> >
> > I have increased my score and removed the ethics concern. The paper would benefit from defining the terminology early on, as it has now been clarified.

---

### Official Review · Reviewer_gPMT · 2025-11-01

**Soundness:** 3
**Presentation:** 3
**Contribution:** 3
**Rating:** 6
**Confidence:** 2

**Summary:**

This paper tackles the problem of predicting the cardinality of object sets and proposes a method that utilize kernel based continuous field that can map to the finite set of spatial objects. In this case, the total density mass is the number of objects. The proposed CORDs provides an invertible mapping that enable the exact decoding from the field space to the discrete space. The authors demonstrate the effectiveness of proposed representation in various applications, including molecular generation, object detection, simulation-based inference and synthetic functions.

**Strengths:**

- The proposed method demonstrates a clean and clear idea that enable the encoding to continuous fields and exact decoding to discrete sets, which also makes the prediction of object cardinality feasible
- This problem it tackles itself is a rather fundamental issue that can be further applied to many downstream tasks as shown in the paper without being restricted to specific use case
- The bijective construction make the transition between discrete and continuous space smooth without the need of additional module

**Weaknesses:**

- The computational cost including the encoding and decoding process might increase when dealing with larger samples
- The experimental performance of CORDS may fall behind some baselines, for instance, the performance in the molecular tasks, so what could be the benefit of this method compared to those baselines in this case.

**Questions:**

- Now, the current representation is rather unified for all downstream tasks, do you think it might be valuable to calibrate current method to become some task-specific representation?
- How the kernel choice and kernel parameter affect the construction?

---

> ### Author Response · Authors · 2025-11-19
>
> The reviewer appreciates the clarity of the CORDS construction and its invertible mapping between discrete sets and continuous fields, but raises three main concerns: (i) encoding and decoding may become costly for larger samples, (ii) CORDS underperforms some strong baselines in molecular tasks, and (iii) it is unclear whether the representation should be task-specific or how sensitive it is to kernel choices.
>
> ---
>
> We thank the reviewer for the detailed feedback. The points raised helped us refine how we present the computational aspects of CORDS and its role as a general-purpose representation. Below we clarify each issue.
>
> ---
>
> **Computational cost of encoding and decoding**
> The encoding cost scales with the number of sampled field points $S$, but in our setup this is not a bottleneck.
>
> Formally,
> $$
> T_{\text{encode}} = O(BSN(d + C)),
> $$
> where $B$ is batch size, $N$ is the number of atoms, $d=3$ for molecules, and $C \lesssim 16$ is the feature dimension. This reduces to one GPU `cdist` computation plus a few reductions. These operations are negligible compared to the cost of a large transformer or diffusion backbone.
>
> With $S \approx 10^3$ and $N \lesssim 30$, the encoding step is well below 1 percent of training time. The training loop never decodes, so decoding cost does not affect scalability. Decoding is only applied at inference to recover discrete molecules and adds a small overhead relative to the sampler.
>
> ---
>
> **Benefit vs stronger molecular baselines**
> Some equivariant GNNs achieve stronger molecular metrics, but our main motivation is to validate a representation that:
>
> 1) models cardinality via density mass,
> 2) supports exact, invertible decoding of positions and continuous features,
> 3) works uniformly across tasks such as molecules, detection, SBI, and synthetic functions.
>
> An additional benefit is that CORDS can model the conditional distribution $p(N \mid c)$ without ever being told the correct cardinality. This follows from the fact that cardinality is encoded directly as integrated density. We do include simple conditional generation experiments, and we agree that richer conditional setups would further highlight this property.
>
> A natural testbed would be conditional protein generation or inpainting, where one should not specify the number of residues in advance (as in ProxelGen). CORDS is well matched to this setting because $N$ is inferred from the field. We did not include protein experiments because they require domain-specific preprocessing and evaluation and would constitute a separate project on their own. We will try to include these experiments for the camera-ready version of the paper.
>
> ---
>
> **Q1: Should the method be calibrated for specific tasks?**
> Yes. For molecular tasks, using an $E(n)$-equivariant backbone is the most direct improvement, since all strong baselines benefit from equivariance. Given that CORDS is already competitive without it, combining the two is promising.
>
> For object detection, the main challenge is overlapping objects. Since all objects contribute to the same density field, identifiability can suffer. Extensions such as using multiple density channels can reduce this issue while keeping the core representation unchanged.
>
> ---
>
> **Q2: Effect of kernel choice and bandwidth**
> The method is not very sensitive to the kernel family. Gaussian, Laplacian, and Epanechnikov kernels all perform similarly. We use the Gaussian kernel for convenience.
>
> Kernel bandwidth $\sigma$ matters more:
>
> - very small $\sigma$ produces sharp kernels and makes decoding sensitive to sampling noise,
> - very large $\sigma$ causes excessive overlap and makes centers harder to recover.
>
> For molecules we found a stable range
> $$
> \sigma \in [0.1\,0.3]
> $$
> (in normalized units). Outside this interval the issues above become more noticeable. We will add a short ablation in the appendix.
>
> ---
>
> We thank the reviewer again for the helpful comments and will incorporate these clarifications in the revised version.
>
> **References**
> Faltings et al., *ProxelGen: Generating Proteins as 3D Densities*

---

### Official Review · Reviewer_6aGq · 2025-11-03

**Soundness:** 4
**Presentation:** 4
**Contribution:** 3
**Rating:** 8
**Confidence:** 3

**Summary:**

The paper proposes to represent discrete sets of arbitrary cardinality as continuous density and feature fields. This enables models to predict/generate sets of objects without fixing cardinality in advance. The authors formally show this bijection. The density field is simply a sum of Gaussians centered at each object, whereas the feature field is the same except each Gaussian is weighted by the corresponding feature vector. Recovering a discrete set requires drawing samples to integrate the density field to find the cardinality $N$. Then, positions are recovered by fitting a Gaussian mixture model to the samples and densities and refining object positions with LBFGS. Feature vectors are recovered by solving a linear system with coefficients and constraints defined by the recovered Gaussians and feature field. The framework of CORDS generalizes Gaussians to generic kernels. CORDS is validated with experiments across 3D molecules, images, astrophysical signals, and local maxima of functions.

**Strengths:**

Motivation for modelling cardinality is clear in the introduction.

Contextualization with related works is extensive without being wordy.

The paper is beautifully written with both intuitive descriptions as well as formalism. Figures are clear and visually appealing.

The proposed method is validated with experiments across multiple domains, showcasing the generality of CORDS. In 3D molecules, CORDS achieves state-of-the art generation quality out of field/voxel-based methods.

**Weaknesses:**

The paper does not discuss its limitations enough. Representing 3D molecules as continuous fields still requires handling discrete samples as inputs to the neural network, so this blows up $N\approx 20$ atoms into $M\approx 1000$ points, and incurs extra cost in integrating densities, gradient-based recovery of positions, and a $O(N^3)$ linear solve to recover feature vectors. It is unclear how much overhead is added by each of these steps, as wall-clock timing, number of training steps, and compute resources are not reported for molecule generation.

**Questions:**

1. In 3D molecule generation, how much is the computational cost of integration-based decoding, both in absolute wall-clock timing and relative to the neural network part? How long does training take, in wall-clock time and number of training steps, and what compute resources are used?
2. Is it possible to make the kernel learnable?
3. Why is it that (line 1129) "feature solve typically dominates cost only for very small $\hat{N}$"?
4. The bijection between discrete sets and continuous fields seems more promising in the other direction: mapping high-dimensional signals down to sparse discrete sets, like tokenization. Alternatively, one could coarse-grain large, discrete sets into continuous fields with fewer points. Does the CORDS framework enable/motivate any approaches in this line of thinking? In physics and chemistry, there are many continuous fields that if represented as discrete but sparse objects could reap benefits: electron densities, wavefunctions, scalar and vector fields in quantum field theory and statistical field theory (e.g. $\phi^4$ theory or solutions to a Landau-Ginzburg Hamiltonian).
5. Future work in decoding spectra (NMR, IR) with overlapping signals could be interesting.

Some potentially related works are Gaussian splatting for rendering images [1] and representing electron density as sums of Gaussians [2].

[1] Kerbl, B., Kopanas, G., Leimkühler, T., & Drettakis, G. (2023). 3D Gaussian splatting for real-time radiance field rendering. ACM Trans. Graph., 42(4), 139-1.

[2] Elsborg, J., Thiede, L., Aspuru-Guzik, A., Vegge, T., & Bhowmik, A. (2025). ELECTRA: A Cartesian Network for 3D Charge Density Prediction with Floating Orbitals. arXiv preprint arXiv:2503.08305.


nit-picking:
1. line 269: "using Eq. equation 1" has an extra word
2. Several citations do not have parentheses around them.
3. line 886: "we will define Field of Graph is a quadruple"
4. line 1062: BIC is not defined
5. line 1102: there is no mention of Laplacian/Epanechnikov kernels elsewhere in the paper
6. line 1253: "diffusion machinery"
7. When introducing kernels, it may be clearer to suggest that the reader imagine Gaussians as a typical example.

---

> ### Author Response · Authors · 2025-11-19
>
> The reviewer is very positive about the motivation, clarity, and breadth of experiments, and notes that CORDS achieves state-of-the-art results among field/voxel-based methods for 3D molecule generation. The main remaining concern is computational: field representations expand atoms into sampled points, and decoding requires integration, kernel fitting, and a linear solve, whose overhead is not quantified. The reviewer also asks about learnable kernels, the complexity statement in Appendix C.1, possible “reverse” uses of the bijection for tokenization/coarse-graining, and connections to spectra decoding and related work. Due to the strict length limit, we keep answers brief here and will expand in the camera-ready.
>
> ---
>
> **Q1: Computational cost and training setup**
>
> There are two distinct components:
>
> 1. **Encoding during training (inside the gradient loop).**
>    For batch size $B$, up to $N$ atoms, spatial dimension $d$, feature dimension $C$, and $S$ samples, encoding scales as
>    $$
>    T_{\text{encode}} = O\bigl(B S N (d + C)\bigr).
>    $$
>    In our molecular setups ($B \approx 32$, $N \le 30$ on QM9, moderate $N$ on GeomDrugs, $S=2048$), a GPU-parallel PyTorch implementation makes this step add under 1 percent overhead to the per-step wall-clock time compared to the Erwin+EDM backbone. We will report profiler numbers in the revised paper.
>
> 2. **Decoding during inference (outside the gradient loop).**
> 	Training is done entirely in field space; decoding is only used for sampling/evaluation. For a decoded molecule with $\hat N$ atoms and $S$ samples, the feature reconstruction step scales as
> 	$$
> 	T_{\text{features}} = O(S \hat N^2 + \hat N^3),
> 	$$
> 	with $S \gg \hat N$ in our experiments. In practice, however, the most expensive part of decoding is the LBFGS refinement of positions, which repeatedly evaluates kernel-based fields; this step is optional but improves performance. We will re-run our experiments to measure and report the exact percentage of generation time spent in CORDS decoding for QM9 and GeomDrugs, as well as hardware, and other details.
>
>
> ---
>
> **Q2: Learnable kernels**
>
> In principle, kernels can be learnable. In the current work we fix a positive kernel with known mass $\alpha$ to guarantee that (i) cardinality is given by density mass$/\alpha$ and (ii) the Gram matrix in feature inversion is positive-definite.
>
> For tasks that stay in field space (e.g., QM9 regression), learning kernel hyperparameters (such as Gaussian $\sigma$) is straightforward and differentiable, and we view this as a natural extension.
>
> For generative tasks that require decoding, the situation is more subtle: the metrics we care about (validity, stability, etc.) are measured on discrete molecules after a non-differentiable decoding pipeline (rounding, model selection in GMM fitting, LBFGS). Diffusion loss is only an indirect proxy, and does not correspond to the generative quality. So in this case, there is no straightforward way to make the kernel parameters learnable.
>
> ---
>
> **Q3: “Feature solve dominates cost only for very small $\hat N$”**
>
> We agree this phrasing is confusing. For feature reconstruction we have
> $$
> T_{\text{features}} = O(S \hat N^2 + \hat N^3).
> $$
> In our regime $S \gg \hat N$, the $S \hat N^2$ term dominates, and the $\hat N^3$ linear solve is negligible. The intended message was that the additional $\hat N^3$ term becomes relevant only when $\hat N$ is very large and $S$ is small. We will reword this part to make it more clear.
>
> ---
>
> **Q4: Using the bijection “in the other direction”**
>
> We agree this is a very interesting direction. In our current experiments we typically start from $N$ objects and sample fields at $S \gg N$ points. Conceptually, one can invert this picture and view CORDS as mapping a complex field to a sparse set of kernel centers and features, effectively performing tokenization or coarse-graining. This corresponds to $N \gg S$ on the decoding side.
>
> We have thought about such ideas in the context of proteins and macromolecules (compressing many atoms into fewer effective interaction sites), but have not yet explored them experimentally. We will mention this explicitly in the Discussion as a natural extension of the framework.
>
> ---
>
> **Q5: Spectra decoding and related work**
>
> Our Fast Radio Burst experiment was designed precisely as a 1D example of this setting: we decode overlapping transient components from a noisy time series, which is conceptually very close to the spectra problems the reviewer mentions (e.g., NMR, IR).
>
> ---
>
> We thank the reviewer for pointing to 3D Gaussian splatting and ELECTRA. We were aware of Gaussian splatting; the ELECTRA work is new to us and encouraging to see in scientific ML. We believe there is strong potential synergy between their work and CORDS-style invertible fields.
>
> ---
>
> **Minor issues**
>
> We will fix the noted typos and inconsistencie. Again, we keep this response brief due to the 5k-character limit.

---

> > ### Author Response · Authors · 2025-11-26
> > **Computational times**
> >
> > As an additional note, we provide the requested information on compute resources and runtimes. The molecule-generation experiments (QM9 and GeomDrugs) were by far the most computationally demanding, so we focus on them here. Our QM9 models were trained for approximately 5 days on a single H100 GPU, using a batch size of 128, for roughly 700k training steps.
> >
> > Below we summarize the training and inference timings. As shown, CORDS encoding adds almost no overhead, and during inference the CORDS decoding (which consists mainly of position and feature reconstruction) time is on the same order as the generation. We emphasize that this is partly due to our current decoding implementation not being fully vectorized for position fitting; this step dominates decoding time at the moment. We plan to optimize this in the public release and expect around an order-of-magnitude speedup.
> >
> > Finally, we would just like to point out that encoding is used only during training, and decoding is used only during inference. If the reviewer has any additional questions or requests for clarifications, we are happy to provide further details and will incorporate these numbers into the revised manuscript in the coming days!
> >
> >
> > | **Section** | Component                      | Time (ms) per molecule | Notes / Description                                   |
> > |-------------|---------------------------------|-------------------------|--------------------------------------------------------|
> > | **Training**| Forward + backward pass         | 4.32                   | Based on measured per-step timing on H100         |
> > |             | Encode                          | 0.014                  | Molecule → CORDS field representation                    |
> > |             |                                 |                         |                                                        |
> > | **Inference**| Neural network sampling        | 58.8                   | Generation (30 steps) on H100                          |
> > |             | Decode: Position reconstr.      | 19.98                  | GMM search + LBFGS refinement                          |
> > |             | Decode: Feature recovery        | 0.57                   | Atom-type / feature reconstruction                     |
> > |             | **Decode total**                | **20.8**               | Integration-based decoding (all parts)                 |
> > |             | **End-to-end total**            | **79.6**               | Sampling + full decoding                               |

---

### Official Review · Reviewer_BLiG · 2025-11-03

**Soundness:** 3
**Presentation:** 3
**Contribution:** 3
**Rating:** 6
**Confidence:** 3

**Summary:**

This paper introduces CORDS, a framework that maps variable-sized discrete sets (e.g., points with features) into continuous representations through density and feature fields. Each element contributes a kernel function centered at its position, whose integrated density encodes cardinality while the feature field carries attribute information. The author conducts experiments on diverse domains—molecular generation, object detection (MultiMNIST), scientific simulation inference, and detection of local maxima. Experiments show competitive performance and robustness to variable object counts.

**Strengths:**

CORDS provides an elegant formulation that continuously encodes cardinality, positions, and attributes in a single differentiable field, with provable invertibility. This offers a strong conceptual link between discrete structures and continuous implicit fields.

The framework is tested across four distinct domains, showing broad potential beyond a single application type.

**Weaknesses:**

The approach relies on dense sampling (∼10³ points per structure) for high-fidelity reconstruction, which can become prohibitively expensive for large-scale data (e.g., complex molecules or real images). The paper lacks quantitative analysis of runtime, memory footprint, or scaling trends with respect to the number of objects or spatial resolution. Without this, the practical feasibility on larger datasets remains unclear.

While CORDS achieves performance comparable to existing models on molecular generation task, it does not clearly surpass them in generation quality. Moreover, as the number of atoms increases, the computational overhead of the continuous-field representation (especially the sampling and kernel-matching stages) grows rapidly, potentially making the approach less scalable than discrete or equivariant diffusion models. Given these results, it remains unclear what practical advantage CORDS offers for molecular generation compared to current SOTA methods—beyond the theoretical elegance of its continuous formulation.

How are weights for LMSE and count penalty (λ) tuned in detection, and how do they affect OOD-count robustness?

**Questions:**

See Weaknesses.

---

> ### Author Response · Authors · 2025-11-19
>
> The reviewer appreciates the conceptual elegance of CORDS and its broad experiments, but raises three concerns: (1) CORDS relies on dense sampling (≈$10^3$ points per structure) and the paper does not analyze runtime, memory, or scaling; (2) in molecular generation, CORDS is competitive but does not clearly surpass strong equivariant baselines, so its practical advantage is unclear; (3) for detection, the reviewer asks how the LMSE and count-penalty weights are chosen and how they affect robustness to out-of-distribution (OOD) object counts.
>
> ---
>
> We thank the reviewer for this constructive feedback. Below we clarify the intended scope of CORDS and how we plan to strengthen the discussion of scalability, molecular benefits, and detection losses.
>
> ---
>
> **Q1: Scalability, dense sampling, and large-scale data**
>
> In short, we use moderate sampling budgets ($1024/2048$ points), Erwin scales linearly and is built for much larger point sets, and we will add an explicit scaling study; like existing GNN-based methods, we do not currently target systems with many thousands of atoms or extreme crowd-counting scenes.
>
> On QM9 (≤$27$ atoms) we use $1024$ points, and on GEOM-DRUGS (>$100$ atoms) $2048$ points. This is a tunable hyperparameter: more points improve reconstruction, fewer points reduce cost. In practice, these settings did not dominate runtime. Erwin’s complexity is linear in the number of sampled points and it has been demonstrated on up to tens of thousands of points, so our regimes are well within its design range. In the revision, we will add a small scaling plot varying the number of samples and reporting runtime, memory, and performance.
>
> For macromolecules/proteins with thousands of atoms, we agree that a direct application becomes expensive; this is also a regime where many GNN-based and equivariant diffusion models struggle. On the other hand, for images, cost scales with the number of *objects*, not pixels: once objects are identified, we can encode them by importance sampling around their locations. For very dense crowd-counting with hundreds or thousands of objects, our current decoding would indeed be prohibitive, and we will flag this as a limitation.
>
> ---
>
> **Q2: Practical advantage of CORDS vs existing molecular generation methods**
>
> The main practical advantage of CORDS is the natural modeling of the *distribution over the number of objects* via density mass. We agree that on QM9 CORDS does not yet clearly outperform strong equivariant baselines and will state this plainly.
>
> Because cardinality is encoded as density mass, the model can flexibly represent and extrapolate $p(N \mid c)$ without an explicit discrete head over $N$. This is what we aim to highlight with the conditional generation experiment that removes a property bin: even when an entire conditioning interval is unseen, the learned density still yields stable conditional count distributions. The same mechanism underlies our OOD-count behavior in detection.
>
> The invertible field–set mapping also gives a clean route from edited fields back to discrete molecules (including continuous features), which is attractive for inpainting or partial conditioning. We have not yet explored protein-scale conditional tasks with CORDS, mainly because suitable datasets and evaluation protocols are non-trivial and would likely require a dedicated follow-up study; we will clarify this to avoid over-claiming.
>
> ---
>
> **Q3: LMSE and count-penalty weights, and OOD-count robustness**
>
> We use simple, lightly tuned weights that emphasize localization/box size, while the count term gently calibrates density mass:
>
> - $w_\text{cls} = 1$ (classification fields),
> - $w_\rho = 4$ (density field $\rho$),
> - $w_\text{size} = 10$ (box size fields),
> - $w_\text{count} = 0.05$ (count penalty on total density mass).
>
> $\rho$ and size carry most of the supervision signal for AP, so we weight them higher; the count term is intentionally small so it corrects systematic over/under-counting without dominating the pixel-wise loss. These values were chosen once to balance gradient magnitudes rather than through extensive tuning.
>
> OOD-count robustness mainly comes from representing counts as density mass (no fixed query budget or hard $N$) plus this light calibration. In the revised version, we will add a short ablation varying $w_\text{count}$ (including setting it to zero) and report effects on in-distribution and OOD count errors and AP.

---

### Official Review · Reviewer_YEsV · 2025-11-03

**Soundness:** 2
**Presentation:** 3
**Contribution:** 2
**Rating:** 4
**Confidence:** 3

**Summary:**

The paper proposes a representation for discrete data based on neural fields, named CORDS. The central idea is to define an invertible mapping between features and continuous fields. The authors show experiments with CORDS on different tasks/domains (molecular generation, object detection, simulation-based inference in astronomy).

**Strengths:**

* The unification of continuous fields and discrete encodings is interesting.
* The presentation is generally clear, and the theoretical derivations are also clear.
* The idea is applied to different tasks/domains, which means that the method shows some generalization.

**Weaknesses:**

* In molecular generation (Table 1) the performance is considerably lower than other models. In molecular regression (Table 4) the performance is also significantly worse than other methods (e.g. EGNN, PaiNN, SE-GNN). In particular, the dipole estimation has the worst error (0.086) of all methods, several times higher than other methods (e.g. PaiNN: 0.012, SE-GNN: 0.023). Results for object detection are limited and have a similar issue. This does not match the "competitive performance" described in the abstract. Results for OOD are relatively better, but generally worse than other methods.

**Questions:**

* The idea seems interesting, but the method is not showing particularly good results for the chosen tasks. Moreover, the results do not support the initial claims.

* For molecular benchmarks, please clarify if baseline numbers were re-run or taken from earlier papers (if any).

---

> ### Author Response · Authors · 2025-11-19
>
> The reviewer appreciates the core idea but notes that (i) molecular generation and regression lag behind strong equivariant GNN baselines, (ii) object detection results do not clearly support the “competitive performance” phrasing, and (iii) the experiments may not fully support our initial claims. They also ask whether the baseline numbers in Tables 1 and 4 were re-run or taken from prior work.
>
> ---
>
> **Q1: Do the empirical results really support “competitive performance”?**
>
> We did not intend “competitive” to suggest matching state-of-the-art equivariant GNNs. Our aim was to show that a single, unified representation can work across domains with minimal tuning. In this sense, CORDS reaches the general ballpark of strong GNN-based methods while outperforming existing voxel-based and continuous representations. We will state this more clearly in the revised version of the paper.
>
> - *Molecular generation vs. equivariant GNNs.*
>   Table 1 compares CORDS to strong E(3)-equivariant models such as EDM, GeoLDM, Rapidash, and Ponita. These methods rely heavily on geometric inductive biases. Since CORDS uses a non-equivariant transformer backbone, we do not expect to surpass them. The intended message is that performance is reasonably close despite the lack of equivariance and the fact that the same architecture is used across multiple domains.
>
> - *Continuous / voxel-based baselines.*
>   In Figure 3, evaluated under the VoxMol/FuncMol OpenBabel protocol, CORDS matches or exceeds these baselines (e.g., higher molecule stability and uniqueness). This is the context in which we meant “competitive performance,” and we will update the text to reflect this more clearly.
>
> - *Object detection and OOD robustness.*
>   In the MultiMNIST setting, CORDS is close to DETR/YOLO in-distribution and shows the smallest relative drop under out-of-distribution object counts. This follows from representing cardinality as density mass rather than using fixed slots or grids. We will rephrase our claim as: “comparable in-distribution performance with improved robustness under OOD cardinality.”
>
> Finally, we avoided using domain-specific architectural tweaks. Adding such components (e.g., equivariant layers for molecules) is natural future work and would likely narrow performance gaps.
>
> ---
>
> **Q2: Were the molecular baselines re-run or taken from previous papers?**
>
> All baseline results were taken directly from the literature; only CORDS was re-run.
>
> - *Table 1 (QM9, GeomDrugs).*
>   Baselines adapted from the GeoLDM paper and related molecular generation works, using their RDKit evaluation protocol.
>
> - *Figure 3 (QM9 continuous/voxel baselines).*
>   VoxMol and FuncMol metrics taken directly from the FuncMol paper under their OpenBabel evaluation.
>
> - *Table 4 (QM9 regression).*
>   Baselines such as DimeNet++, EGNN, PaiNN, and SEGNN taken from the paper *E(n) Equivariant Graph Neural Networks*, we use the reported MAEs.
>
> In the final version we will clearly mark which results come from prior work, cite the original sources in captions, and specify which evaluation protocol (RDKit vs. OpenBabel) each table uses.
>
> ---
>
> **References**
> Xu et al., *Geometric Latent Diffusion Models for 3D Molecule Generation*
> Pinheiro et al., *3D molecule generation by denoising voxel grids*
> Kirchmeyer et al., *Score-based 3D molecule generation with neural fields*
> Satorras et al., *E(n) Equivariant Graph Neural Networks*

---

### Author Response · Authors · 2025-11-19

We thank reviewers `YEsV`, `BLiG`, `6aGq`, `gPMT`, and `7wBB` for their time and detailed feedback. We are encouraged that several reviewers are receptive to the core idea: `6aGq` highlights the clear motivation, concise yet extensive related-work discussion, and the balance between intuitive descriptions and formalism, and notes that CORDS achieves state-of-the-art generation quality among field/voxel-based methods for 3D molecules, while `BLiG` and `gPMT` emphasize the elegance and generality of the bijection between sets and fields across multiple domains.

Our read of the main points in each review is:

- `YEsV`: Values the unification of continuous fields and discrete encodings, clear presentation, and multi-domain experiments. Main concern: molecular generation and regression lag behind strong baselines, so the abstract’s “competitive performance” sounds too strong. Also asks how baselines were obtained and whether results fully support the claims.

- `BLiG`: Highlights CORDS as an elegant invertible field representation that jointly encodes cardinality, positions, and attributes, with promising results on four domains. Main issues: reliance on dense sampling, lack of runtime/memory/scaling analysis, and no clear advantage over existing molecular models in current experiments.

- `6aGq`: Very positive about the motivation for modeling cardinality, the balance of intuition and formalism, and the wide range of applications, and notes that CORDS achieves state-of-the-art generation quality among field/voxel-based methods on 3D molecules. Main concern: limitations and overheads (sampling, integration, position refinement, feature solves) are not discussed or quantified in enough detail.

- `gPMT`: Stresses that CORDS gives a clean, invertible construction for cardinality prediction with broad potential. Concerns: encoding/decoding cost may become large for bigger samples, and molecular performance can lag behind baselines; asks about task-specific variants and sensitivity to kernel choices.

- `7wBB`: Criticizes the abstract/introduction for using non-standard terms (“variable cardinality”, “bijective representation”, “feature field”, etc.) before defining them, and asks what precise ML problem is being solved. Also raises an ethics concern about AI reliance; we note that the paper already includes an explicit LLM usage statement clarifying that LLMs were used only for polishing text and helping reconstruct baselines, not for ideas, theory, or results.

We respond to individual points below. Here we briefly highlight two recurring themes:

- **Empirical performance and “competitive” wording (`YEsV`, `BLiG`, `gPMT`)**
  These reviewers note that our molecular results do not match the strongest specialized equivariant GNNs and that “competitive performance” can be read too strongly. We agree that CORDS does not claim SOTA on these benchmarks. Our goal is to show that a single, unified field-based representation stays in the general performance range of strong graph methods, improves over prior continuous/voxel baselines, and naturally models both object features and cardinality. We will revise the abstract and main text to make this representation-first, cross-domain focus explicit and soften the wording around “competitive”.

- **Scalability and computational cost (`BLiG`, `6aGq`, `gPMT`)**
  These reviewers highlight that CORDS uses more samples than graphs and introduces additional decoding overhead. We agree that scalability is an important limitation that merits clearer quantification. Conceptually, encoding runs inside the gradient loop, is fully GPU-parallelized, and in our current setups contributes only a negligible fraction of per-step training cost relative to the Erwin+EDM backbone. Decoding, by contrast, is used only at inference and is where most of the additional cost arises, primarily due to position refinement and feature reconstruction. While graph-based models often incur quadratic costs in the number of nodes, CORDS can partially offset its sampling overhead by pairing the field representation with sparse-attention transformers such as Erwin, which scale close to linearly in the number of sampled points. To address the reviewers’ requests, we now provide concrete runtime measurements in Appendix C, including a table summarizing the computational cost of CORDS encoding and decoding relative to standard training and inference components in our molecular-generation experiments.

We hope this overall comment helps contextualize our more detailed responses, and we again thank the reviewers and the area chair for their thoughtful feedback. We have uploaded a revised version of the manuscript with all changes suggested by reviewers clearly highlighted in blue.

---

### Meta-Review · Area_Chair_FuWN · 2026-01-06

**Summary:**

This paper proposes CORDS, a framework that represents variable-cardinality discrete structures as continuous density and feature fields with an invertible mapping back to discrete sets. The idea is technically elegant, clearly formalized, and evaluated across a broad range of tasks, including molecular generation, object detection, and scientific inference.

Among the five reviewers, only reviewer 7wBB followed up after the author rebuttal. The other four reviewers did not respond further. Reviewer 7wBB indicated that they increased their score after the rebuttal, although the updated numerical score is not visible. After carefully reading the paper and checking the rebuttal, I believe the authors adequately addressed the substantive concerns raised by the other reviewers, even though those reviewers did not explicitly follow up. Overall, the paper makes a strong and general contribution and should be accepted.

**Reviewer Concerns:**

The main concerns raised across reviews related to (i) empirical performance relative to strong equivariant molecular baselines, (ii) computational cost and scalability of the continuous field representation, and (iii) clarity of terminology and problem formulation, particularly in the abstract and introduction.

Although most reviewers did not reply after the rebuttal, I carefully checked the authors’ responses and the additional clarifications provided. The authors clearly explained that CORDS does not aim to outperform specialized equivariant models on molecular benchmarks, but instead to offer a unified representation that naturally handles unknown cardinality across domains. They also provided detailed runtime measurements, scalability discussion, and clarified where encoding and decoding costs arise. I find these responses sufficient to address concerns about practicality and over-claiming.

Regarding reviewer 7wBB, their original review focused primarily on terminology and phrasing, with limited engagement with the main technical sections where the mathematical definitions are provided. After the rebuttal clarified these definitions and addressed the ethics concern, the reviewer explicitly stated that they raised their score and removed the ethics flag. I agree that the remaining issues raised by this reviewer are largely resolved through clarification rather than substantive technical changes.

Overall, after reading the paper and rebuttal in full, I find that the key concerns have been addressed and that the remaining limitations are clearly acknowledged by the authors.

**Reviewer Scores:**

Before the rebuttal, the reviewer scores included 2, 4, 6, 6, and 8; 2 was from reviewer 7wBB. After the rebuttal, reviewer 7wBB explicitly stated that they increased their score and removed the ethics concern, although the final numeric score is not shown and is expected to be greater than or equal to 4. The other four reviewers did not follow up, and therefore no reviewer explicitly indicated an intention to change their score following the rebuttal or discussion.

---

### Decision · Program_Chairs · 2026-01-26

Accept (Poster)